# Theoretically Understanding Data Reconstruction Leakage in Federated Learning

## Abstract

Federated learning (FL) is an emerging collaborative learning paradigm that aims to protect data privacy. Unfortunately, recent works show FL algorithms are vulnerable to the serious data reconstruction attacks, and a series of follow-up works are proposed to enhance the attack effectiveness. However, existing works lack a theoretical foundation on to what extent the devices' data can be reconstructed and the effectiveness of these attacks cannot be compared fairly due to their unstable performance. To address this deficiency, we propose a theoretical framework to understand data reconstruction attacks to FL. Our framework involves bounding the data reconstruction error and an attack's error bound reflects its inherent attack effectiveness. Under the framework, we can theoretically compare the effectiveness of existing attacks. For instance, our results on multiple datasets validate that the iDLG data reconstruction attack inherently outperforms the DLG attack.

## 1 Introduction

The emerging collaborative data analysis using federated learning (FL) (McMahan et al., 2017) has been a great potential to protect data privacy. In FL, the participating devices keep and train their data locally, and only share the trained models (e.g., gradients or parameters), instead of the raw data, with a center server (e.g., cloud). The server updates its global model by aggregating the received device models, and broadcasts the updated global model to all participating devices such that all devices *indirectly* use all data from other devices. FL has been deployed by many companies such as Google Federated Learning (2022), Microsoft Federated Learning (2022), IBM Federated Learning (2022), and Alibaba Federated Learning (2022), and applied in various privacy-sensitive applications, including on-device item ranking (McMahan et al., 2017), content suggestions for on-device keyboards (Bonawitz et al., 2019), next word prediction (Li et al., 2020a), health monitoring (Rieke et al., 2020), and medical imaging (Kaissis et al., 2020).

Unfortunately, recent works show that, although only sharing device models, it is still possible for an adversary (e.g., the malicious server) to perform the severe *data reconstruction attack* to FL (Zhu et al., 2019), where an adversary could *directly* reconstruct the device's training data via the shared device models. Later, a bunch of follow-up enhanced attacks (e.g, Hitaj et al. (2017); Wang et al. (2019); Zhao et al. (2020); Wei et al. (2020); Yin et al. (2021); Jeon et al. (2021); Zhu & Blaschko (2021); Dang et al. (2021); Balunovic et al. (2022); Li et al. (2022); Fowl et al. (2022); Wen et al. (2022); Haim et al. (2022)) are proposed by either incorporating some (known or unrealistic) prior knowledge or requiring an auxiliary dataset to simulate the training data distribution.

However, we note that existing attack methods have several limitations: First, they are sensitive to the initialization. For instance, we show in Figure 1 that the attack performance of iDLG (Zhao et al., 2020) and DLG (Zhu et al., 2019) are significantly influenced by initial parameters (i.e., the mean and standard deviation) of a Gaussian distribution, where the initial data is sampled from. Second, existing attack methods mainly show comparison results on a FL model at a snapshot, which cannot reflect

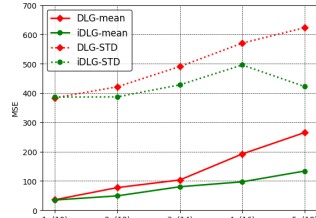

Figure 1: Impact of the initial parameters of a Gaussian distribution on the data reconstruction attack performance. a (b) in the x-axis indicates the mean (standard deviation) of the Gaussian. The default mean and standard deviation are both 1.

attacks' true effectiveness. As FL training is dynamic, an adversary can perform attacks in any stage of the training. Hence, Attack A shows better performance than Attack B at a snapshot does not imply A is truly more effective than B. Third, worse still, they lack a theoretical understanding on to what extent the training data can be reconstructed. These limitations make existing attacks cannot be easily and fairly compared and hence it is unknown which attacks are inherently more effective.

In this paper, we would like to ask the questions: is it possible to measure the effectiveness of data reconstruction attacks to FL from a theoretical perspective? Also, can we theoretically compare the existing data reconstruction attacks at any stage of training? The answers are "Yes", under certain (mild) assumptions. Specifically, we propose a theoretical framework to understand data reconstruction attacks to FL. Our framework aims to bound the error between the private data and the reconstructed counterpart in the whole FL training, where an attack's (smaller) error bound reflects its inherent (better) attack effectiveness. Our theoretical results show that when an attacker's reconstruction function has a smaller Lipschitz constant, then this attack intrinsically performs better. Under the framework, we can theoretically compare the existing attacks by directly comparing their bounded errors. We test our framework on state-of-the-art attacks and multiple benchmark datasets. For example, our experimental results show that InvGrad (Geiping et al., 2020) performs better than both DLG (Zhu et al., 2019) and iDLG (Zhao et al., 2020) on complex datasets, while iDLG is comparable or performs slightly better than DLG.

## 2 RELATED WORK

Existing data reconstruction attacks to FL are classified as optimization-based and close-form based.

**Optimization-based data reconstruction attacks to FL:** A series of works (Hitaj et al., 2017; Zhu et al., 2019; Wang et al., 2019; Zhao et al., 2020; Wei et al., 2020; Yin et al., 2021; Jeon et al., 2021; Dang et al., 2021; Balunovic et al., 2022; Fowl et al., 2022; Wen et al., 2022; Li et al., 2022) formulate data reconstruction attacks as the *gradient matching* problem, i.e., an optimization problem that minimizes the difference between gradient from the raw data and that from the reconstructed counterpart. Some works found the gradient itself includes insufficient information to well recover the data (Jeon et al., 2021; Zhu & Blaschko, 2021). For example, Zhu & Blaschko (2021) show there exist pairs of data (called twin data) that visualize different, but have the same gradient. To mitigate this issue, a few works propose to incorporate prior knowledge (e.g., total variation (TV) regularization (Geiping et al., 2020; Yin et al., 2021), batch normalization (BN) statistics (Yin et al., 2021)) into the training data, or introduce an auxiliary dataset to simulate the training data distribution (Hitaj et al., 2017; Wang et al., 2019; Jeon et al., 2021) (e.g., via generative adversarial networks (GANs) Goodfellow et al. (2014)). Though empirically effective, these methods are less practical or data inefficient, e.g., TV is limited to natural images, BN statistics are often unavailable, and training an extra model requires a large amount of data samples.

**Closed-form data reconstruction attacks to FL:** A few recent works (Geiping et al., 2020; Zhu & Blaschko, 2021; Fowl et al., 2022) derive closed-form solutions to reconstruct data, but they require the neural networks used in the FL algorithm are fully connected (Geiping et al., 2020), linear/ReLU Fowl et al. (2022), or convolutional (Zhu & Blaschko, 2021).

We will design a framework to theoretically understand the data reconstruction attack to FL in a general setting, and provide a way to compare the effectiveness of the existing attacks.

## 3 PRELIMINARIES AND PROBLEM SETUP

### 3.1 FEDERATED LEARNING (FL)

**Objective function:** The FL paradigm enables a server to coordinate the training of multiple local devices through multiple rounds of global communications, without sharing the local data. Suppose there are $N$ devices and a central server participating in FL. Each $k$-th device owns a training dataset $D^k = \{(\mathbf{x}_j^k, y_j^k)\}_{j=1}^{n_k}$ with $n_k$ samples, and each sample $\mathbf{x}_j^k$ has a label $y_j^k$. FL considers the following distributed optimization problem:

$$\min_{\mathbf{w}} \mathcal{L}(\mathbf{w}) = \sum_{k=1}^{N} p_k \mathcal{L}_k(\mathbf{w}), \tag{1}$$

where $p_k \geq 0$ is the weight of the $k$-th device and $\sum_{k=1}^{N} p_k = 1$; the $k$-th device's local objective is defined by $\mathcal{L}_k(\mathbf{w}) = \frac{1}{n_k} \sum_{j=1}^{n_k} \ell(\mathbf{w}; (\mathbf{x}_j^k, y_j^k))$, with $\ell(\cdot; \cdot)$ an algorithm-dependent loss function.

**FedAvg** McMahan et al. (2017): It is the *de factor* FL algorithm to solve Equation (1) in an iterative way. In each communication round, it only shares the gradients $\nabla_{\mathbf{w}}\mathcal{L}_k(\mathbf{w})$ instead of the original data $D^k$ to the server for each $k$-th device. Specifically, in the current round $t$, each $k$-th device first downloads the latest global model (denoted as $\mathbf{w}_{t-1}$) from the server and initializes its local model as $\mathbf{w}_t^k = \mathbf{w}_{t-1}$; then it performs (e.g., $E$) local SGD updates as below:

$$\mathbf{w}_{t+j}^k \leftarrow \mathbf{w}_{t+j-1}^k - \eta_{t+j}\nabla\mathcal{L}_i(\mathbf{w}_{t+j}^k; \xi_{t+j}^k), \, j = 1, 2, \cdots, E, \tag{2}$$

where $\eta_{t+j}$ is the learning rate and $\xi_{t+j}^k$ is sampled from the local data $D^k$ uniformly at random.

Next, the server updates the global model $\mathbf{w}_t$ for the next round by aggregating either full or partial device models. The final global model is downloaded by all devices for their learning tasks.

- *Full device participation.* It requires all device models used for aggregation, and the server performs $\mathbf{w}_t \leftarrow \sum_{k=1}^N p_k \mathbf{w}_t^k$ with $p_k = \frac{n_k}{\sum_{i=1}^N n_i}$ and $\mathbf{w}_t^k = \mathbf{w}_{t+E}^k$. Note that full device participation means the server must wait for the slowest devices, which is often unrealistic in practice.

- *Partial device participation.* This is a more realistic setting as it does not require the server to know all device models. Suppose the server only needs $K$ ($< N$) device models for aggregation and discards the remaining ones. Let $\mathcal{S}_t$ be the set of $K$ chosen devices in the $t$-th iteration. Then, the server's aggregation step performs $\mathbf{w}_t \leftarrow \frac{N}{K}\sum_{k\in\mathcal{S}_t} p_k \mathbf{w}_t^k$ with $\mathbf{w}_t^k = \mathbf{w}_{t+E}^k$.

**Quantifying the degree of non-iid (heterogeneity):** Real-world FL applications often do not satisfy the iid assumption for data among local devices. Li et al. (2020b) proposed a way to quantify the degree of non-iid. Specifically, let $\mathcal{L}^*$ and $\mathcal{L}_k^*$ be the minimum values of $\mathcal{L}$ and $\mathcal{L}_k$, respectively. Then, the term $\Gamma = \mathcal{L}^* - \sum_{k=1}^N p_k\mathcal{L}_k^*$ is used for quantifying the degree of non-iid. If the data are iid, then $\Gamma$ obviously goes to zero as the number of samples grows. If the data are non-iid, then $\Gamma$ is nonzero, and its magnitude reflects the heterogeneity of the data distribution.

**Assumptions on the devices' loss function:** To ensure FedAvg guarantees to converge to the global optimal, existing works have the following assumptions on the local devices' loss functions $\{\mathcal{L}_k\}$.

**Assumption 1.** $\{\mathcal{L}_k\}'s$ are *L-smooth:* $\mathcal{L}_k(\mathbf{v}) \le \mathcal{L}_k(\mathbf{w}) + (\mathbf{v}-\mathbf{w})^T\nabla\mathcal{L}_k(\mathbf{w}) + \frac{L}{2}\|\mathbf{v}-\mathbf{w}\|_2^2, \forall\mathbf{v}, \mathbf{w}$.

**Assumption 2.** $\{\mathcal{L}_k\}'s$ are *$\mu$-strongly convex:* $\mathcal{L}_k(\mathbf{v}) \ge \mathcal{L}_k(\mathbf{w}) + (\mathbf{v}-\mathbf{w})^T\nabla\mathcal{L}_k(\mathbf{w}) + \frac{\mu}{2}\|\mathbf{v}-\mathbf{w}\|_2^2, \forall\mathbf{v}, \mathbf{w}$.

**Assumption 3.** *Let $\xi_t^k$ be sampled from the $k$-th device's data uniformly at random. The variance of stochastic gradients in each device is bounded:* $\mathbb{E}\left\|\nabla\mathcal{L}_k(\mathbf{w}_t^k, \xi_t^k) - \nabla\mathcal{L}_k(\mathbf{w}_t^k)\right\|^2 \le \sigma_k^2, \forall k$.

**Assumption 4.** *The expected squared norm of stochastic gradients is uniformly bounded, i.e.,* $\mathbb{E}\left\|\nabla\mathcal{L}_k(\mathbf{w}_t^k, \xi_t^k)\right\|^2 \le G^2, \forall k, t$.

Note that Assumptions 1 and 2 are generic. Typical examples include regularized linear regression, logistic regression, softmax classifier, and recent convex 2-layer ReLU networks (Pilanci & Ergen, 2020). Assumptions 3 and 4 are used by the existing works (Stich, 2018; Stich et al., 2018; Yu et al., 2019; Li et al., 2020b) to derive the convergence condition of FedAvg to reach the global optimal.

## 3.2 OPTIMIZATION-BASED DATA RECONSTRUCTION ATTACKS ON FL

Existing works assume the honest-but-curious server, i.e., it follows the FL protocol but wants to infer devices' private information. In data reconstruction attacks, the server has access to all device models in all communication rounds and infers devices' private training data. Given the private data $\mathbf{x} \in [0,1]^d$ with label $y$[1], we denote the reconstructed data by a malicious server as $\hat{\mathbf{x}} = \mathcal{R}(\mathbf{w}_t)$, where $\mathcal{R}(\cdot)$ indicates a *data reconstruction function*, and $\mathbf{w}_t$ can be any intermediate server's global model. Modern optimization-based data reconstruction attacks use different $\mathcal{R}(\cdot)$ functions, but are majorly based on *gradient matching*. Specifically, they aim to solve the below optimization problem:

$$(\hat{\mathbf{x}}, \hat{y}) = \mathcal{R}(\mathbf{w}_t) = \arg\min_{(\mathbf{x}'\in[0,1]^d, y')} \mathbb{E}_{(\mathbf{x},y)}[\text{GML}(g_{\mathbf{w}_t}(\mathbf{x}, y), g_{\mathbf{w}_t}(\mathbf{x}', y')) + \lambda\text{Reg}(\mathbf{x}')], \tag{3}$$

where we let the gradient w.r.t. $(\mathbf{x}, y)$ be $g_{\mathbf{w}_t}(\mathbf{x}, y) := \nabla_{\mathbf{w}}\mathcal{L}(\mathbf{w}_t; (\mathbf{x}, y))$ for notation simplicity. $\text{GML}(\cdot, \cdot)$ means the *gradient matching loss* (i.e., the distance between the real gradients and estimated gradients) and $\text{Reg}(\cdot)$ is an auxiliary *regularizer* for the reconstruction. Here, we list $\text{GML}(\cdot, \cdot)$ and $\text{Reg}(\cdot)$ for three representative data reconstruction attacks, and more attack are in Appendix C.1.2.

---

[1]This can be a single data sample or a batch of data samples.

**Algorithm 1** Iterative solvers for optimization-based data reconstruction attacks

**Input:** Model parameters $\mathbf{w}_t$; true gradient $g(\mathbf{x}, y); \eta, \lambda$.
**Output:** Reconstructed data $\hat{\mathbf{x}}$.

1: Initialize dummy input(s) $\mathbf{x}'_0$ and dummy label(s) $y'_0$
2: **for** $i = 0; i < I; i++$ **do**
3:     $\mathrm{L}(\mathbf{x}'_i) = \mathrm{GML}(g_{\mathbf{w}_t}(\mathbf{x}, y), g_{\mathbf{w}_t}(\mathbf{x}'_i, y'_i)) + \lambda\mathrm{Reg}(\mathbf{x}'_i)$
4:     $\mathbf{x}'_{i+1} \leftarrow \mathrm{SGD}(\mathbf{x}'_i; \theta^i) = \mathbf{x}'_i - \eta\nabla_{\mathbf{x}'_i}\mathrm{L}(\mathbf{x}'_i)$
5:     $\mathbf{x}'_{i+1} = \mathrm{Clip}(\mathbf{x}'_{i+1}, 0, 1)$
6: **end for**
7: **return** return $\mathbf{x}'_I$

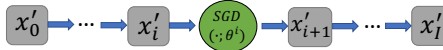

Figure 2: Iterative solvers for data reconstruction attacks as unrolled deep feed-forward networks: We map the $i$-th SGD iteration (parametrized by $\theta^i$) into a single network layer, and stack $I$ layers to form an $I$-layer deep network. Feeding forward data through the $I$-layer network is equivalent to executing $I$ SGD updates. The trainable parameters $\{\theta^i\}$ are colored in blue. These parameters can be learned from real (reconstructed) data by training the deep network in an end-to-end fashion.

- **DLG** (Zhu et al., 2019). It uses the mean squared error as the gradient matching loss, i.e., $\mathrm{GML}(g_{\mathbf{w}_t}(\mathbf{x}, y), g_{\mathbf{w}_t}(\mathbf{x}', y')) = \|g_{\mathbf{w}_t}(\mathbf{x}, y) - g_{\mathbf{w}_t}(\mathbf{x}', y')\|_2^2$ and uses no regularizer.

- **iDLG** (Zhao et al., 2020). It show that the private labels $y$ can be estimated before solving Equation 3. Assuming the estimated label as $\hat{y}$, iDLG solves for $\hat{\mathbf{x}} = \arg\min_{\mathbf{x}'} \mathbb{E}_{\mathbf{x}}[\mathrm{GML}(g_{\mathbf{w}_t}(\mathbf{x}, y), g_{\mathbf{w}_t}(\mathbf{x}', \hat{y})) + \lambda\mathrm{Reg}(\mathbf{x}')]$, where it uses the same $\mathrm{GML}(\cdot)$ as DLG and also has no regularizer.

- **InvGrad** (Geiping et al., 2020). InvGrad improves upon DLG and iDLG. It first estimates the private label as $\hat{y}$ in advance. Then it uses a negative cosine similarity as $\mathrm{GML}(\cdot)$ and a total variation regularizer $\mathrm{Reg}_{\mathrm{TV}}(\cdot)$ as an image prior. Specifically, InvGrad solves for $\hat{\mathbf{x}} = \arg\min_{\mathbf{x}'} \mathbb{E}_{\mathbf{x}}[1 - \frac{\langle g_{\mathbf{w}_t}(\mathbf{x}, y), g_{\mathbf{w}_t}(\mathbf{x}', \hat{y})\rangle}{\|g_{\mathbf{w}_t}(\mathbf{x}, y)\|_2 \cdot \|g_{\mathbf{w}_t}(\mathbf{x}', \hat{y})\|_2} + \lambda\mathrm{Reg}_{\mathrm{TV}}(\mathbf{x}')]$.

Algorithm 1 shows the pseudo-code of iterative solvers for data reconstruction attacks and Algorithm 4 in Appendix shows more details for each attack. Finally, the attack performance is measured by the similarity $\mathrm{sim}(\hat{\mathbf{x}}, \mathbf{x})$ between $\hat{\mathbf{x}}$ and $\mathbf{x}$. The larger similarity, the better attack performance. In the paper, we consider the most common similarity metric, i.e., the negative mean-square-error $\mathrm{sim}(\hat{\mathbf{x}}, \mathbf{x}) = -\mathbb{E}\|\hat{\mathbf{x}} - \mathbf{x}\|^2$, where the expectation $\mathbb{E}$ considers the randomness during reconstruction.

## 4 A THEORETICAL FRAMEWORK TO UNDERSTAND DATE RECONSTRUCTION ATTACKS TO FEDERATED LEARNING

Though many data reconstruction attacks to FL have been proposed, it is still unknown how to *theoretically* compare the effectiveness of existing attacks, as stated in the Introduction. In this section, we will understand data reconstruction attacks to FL from a theoretical perspective. We first derive a reconstruction error bound for convex objective losses, under the Assumptions 1-4. The error bound involves knowing the Lipschitz constant of the data reconstruction function. Directly calculating the exact Lipschitz constant is computationally challenging. We then adapt existing methods to calculate its upper bound. We argue that an attack with a smaller upper bound is intrinsically more effective. We also emphasize that our theoretical framework is applicable to any adversary who knows the global model during FL training (see below Theorems 1 and 2).

### 4.1 BOUNDING THE DATA RECONSTRUCTION ERROR

Give random data $\mathbf{x}$ from a device, our goal is to bound the common norm-based reconstruction error[2], i.e., $\mathbb{E}\|\mathbf{x} - \mathcal{R}(\mathbf{w}_t)\|^2$ at any round $t$, where the $\mathcal{R}(\cdot)$ function can be any existing data reconstruction attack. Directly bounding this error is challenging because the global model dynamically aggregates local device models, which are trained by a (stochastic) learning algorithm and whose learning procedure is hard to characterize during training. To alleviate this issue, we introduce the optimal global model $\mathbf{w}^\star$ that can be learnt by the FL algorithm. Then, we can bound the error as follows:

$$\mathbb{E}\|\mathbf{x} - \mathcal{R}(\mathbf{w}_t)\|^2 = \mathbb{E}\|\mathbf{x} - \mathcal{R}(\mathbf{w}^\star) + \mathcal{R}(\mathbf{w}^\star) - \mathcal{R}(\mathbf{w}_t)\|^2$$
$$\leq 2\big(\mathbb{E}\|\mathbf{x} - \mathcal{R}(\mathbf{w}^\star)\|^2 + \mathbb{E}\|\mathcal{R}(\mathbf{w}^\star) - \mathcal{R}(\mathbf{w}_t)\|^2\big). \quad (4)$$

---

[2]The norm-based mean-square-error (MSE) bound can be easily generalized to the respective PSNR bound. This is because PSNR has a strong connection with MSE, i.e., PSNR = -10 log (MSE). However, the MSE bound is unable to generalize to SSIM or LPIPS since these metrics focus more on image structural information but not pixel differences, and they also do not have an analytic form.

Note that the first term in Equation (4) is a constant and can be directly computed under a given reconstruction function and a strongly convex loss used in FL. Specifically, if the loss in each device is strongly convex, then the global model can converge to the *optimal* $\mathbf{w}^*$ based on (Li et al., 2020b)'s theoretical results. Then we can obtain $\mathcal{R}(\mathbf{w}^*)$ per attack and compute the first term. In our experiments, we run the FL algorithm until the loss difference between two consecutive iterations does not exceed $1e-5$, and treat the resultant global model as $\mathbf{w}^*$.

Now our goal reduces to bounding the second term. However, it is still challenging without knowing any properties of the reconstruction function. To mitigate it, we will have another assumption on the reconstruction function $\mathcal{R}(\cdot)$ as below, which can also be verified in our later sections.

**Assumption 5.** *The existing data reconstruction function $\mathcal{R}(\cdot)$ is $L_{\mathcal{R}}$-Lipschitz continuous: there exists a constant $L_{\mathcal{R}}$ such that $\|\mathcal{R}(\mathbf{v}) - \mathcal{R}(\mathbf{w})\| \leq L_{\mathcal{R}}\|\mathbf{v} - \mathbf{w}\|, \forall \mathbf{v}, \mathbf{w}$.*

The smallest $L_{\mathcal{R}}$ is called the *Lipschitz constant*, which indicates the maximum ratio between variations in the output space and those in the input space. Next, we present our theoretical results. *Note that our error bounds consider all randomness in FL training and data reconstruction, and hence they are the worst-case error under such randomness.*

**Theoretical results with full device participation:** We first analyze the case where all devices participate in the aggregation on the server in each communication round. Assume the **FedAvg** algorithm stops after $T$ iterations and returns $\mathbf{w}_T$ as the solution. Let $L, \mu, \sigma_k, G, L_{\mathcal{R}}$ be defined in Assumptions 1 to 5. Then, we have:

**Theorem 1.** *Let Assumptions 1 to 5 hold. Choose $\gamma = \max\{8L/\mu, E\}$ and the learning rate $\eta_t = \frac{2}{\mu(\gamma+t)}$. Then, for any communication round $t$, FedAvg with full device participation satisfies*

$$\mathbb{E}\|\mathbf{x} - \mathcal{R}(\mathbf{w}_t)\|^2 \leq 2\mathbb{E}\|\mathbf{x} - \mathcal{R}(\mathbf{w}^*)\|^2 + \frac{2L_{\mathcal{R}}^2}{\gamma+t}\Big(\frac{4B}{\mu^2} + (\gamma+1)\mathbb{E}\|\mathbf{w}_1 - \mathbf{w}^*\|^2\Big), \quad (5)$$

*where $B = \sum_{k=1}^{N} p_k^2 \sigma_k^2 + 6L\Gamma + 8(E-1)^2 G^2$.*

**Theoretical results with partial device participation:** As discussed in Section 3, partial device participation is more practical. Recall that $\mathcal{S}_t$ contains the $K$ active devices in the $t$-th iteration. To show our theoretical results, we need to make an assumption on $\mathcal{S}_t$. Specifically, we have the below Assumption 6 stating the $K$ devices are selected from the distribution $p_k$ independently and with replacement, following (Sahu et al., 2018; Li et al., 2020b).

**Assumption 6.** *Assume $\mathcal{S}_t$ includes a subset of $K$ devices randomly selected with replacement according to the probabilities $p_1, \cdots, p_N$. FedAvg performs aggregation as $\mathbf{w}_t \leftarrow \frac{1}{K}\sum_{k\in\mathcal{S}_t}\mathbf{w}_t^k$.*

**Theorem 2.** *Let Assumptions 1 to 6 hold. Let $\gamma, \eta_t,$ and $B$ be defined in Theorem 1, and define $C = \frac{4}{K}E^2G^2$. Then, for any communication round $t$,*

$$\mathbb{E}\|\mathbf{x} - \mathcal{R}(\mathbf{w}_t)\|^2 \leq 2\mathbb{E}\|\mathbf{x} - \mathcal{R}(\mathbf{w}^*)\|^2 + \frac{2L_{\mathcal{R}}^2}{\gamma+t}\Big(\frac{4(B+C)}{\mu^2} + (\gamma+1)\mathbb{E}\|\mathbf{w}_1 - \mathbf{w}^*\|^2\Big). \quad (6)$$

## 4.2 Computing the Lipschitz Constant for Data Reconstruction Functions

In this part, we show how to calculate the Lipschitz constant for the data reconstruction function. Our idea is to first leverage the strong connection between optimizing data reconstruction attacks and the deep neural networks; and then adapt existing methods to approximate the Lipschitz upper bound.

**Iterative solvers for optimization-based data reconstruction attacks as unrolled deep feed-forward networks:** Recent works Chen et al. (2018); Li et al. (2019); Monga et al. (2021) show a strong connection between iterative algorithms and deep network architectures. The general idea is: starting with an abstract iterative algorithm, we map one iteration into a single network layer, and stack a finite number of (e.g., $H$) layers to form a deep network, which is also called *unrolled deep network*. Feeding the data through an $H$-layer network is hence equivalent to executing the iterative algorithm $H$ iterations. The parameters of the unrolled networks are learnt from data by training the network in an end-to-end fashion. From Algorithm 1, we can see that the trajectory of an iterative solver for an optimization-based data reconstruction attack corresponds to a customized unrolled deep feed-forward network. Specifically, we treat $\mathbf{w}_t$ and the initial $\mathbf{x}_0'$ as the input, the intermediate reconstructed $\mathbf{x}_i'$ as the $i$-th hidden layer, followed by a clip nonlinear activation function, and the final reconstructed data $\hat{\mathbf{x}} = \mathbf{x}_I'$ as the output of the network. Given intermediate $\{\mathbf{x}_i'\}$ with a set of

**Algorithm 2** AutoLip

**Input:** function $f$ and its computation graph $(g_1, ..., g_H)$
**Output:** Lipschitz upper bound $L_{AutoLip} \geq L_f$
1: $\phi_0(\mathbf{x}) = \mathbf{x}; \phi_h(\mathbf{x}) = f(\mathbf{x})$
2: $\phi_h(\mathbf{x}) = g_h(\mathbf{x}, \phi_1(\mathbf{x}), \cdots, \phi_{h-1}(\mathbf{x})), \forall h \in [1, H]$
3: $\mathcal{Z} = \{(z_0, ..., z_H) : \forall h \in [0, H], \phi_h \text{ is constant} \Rightarrow z_h = \phi_h(0)\}$
4: $L_0 \leftarrow 1$
5: **for** $h = 1$ to $H$ **do**
6: $\quad L_h \leftarrow \sum_{i=1}^{h-1} \max_{z \in \mathcal{Z}} \|\partial_i g_h(z)\|_2 L_i$
7: **end for**
8: **return** $L_{AutoLip} = L_H$

**Algorithm 3** Power method to calculate the matrix $\ell_2$-norm

**Input:** affine function $f : \mathbf{R}^n \to \mathbf{R}^m$, #iterations $Iter$
**Output:** Upper bound of the Lipschitz constant $L_f$
1: **for** $j = 1$ to $Iter$ **do**
2: $\quad v \leftarrow \nabla g(v)$ where $g(x) = \frac{1}{2}\|f(x) - f(0)\|_2^2$
3: $\quad \lambda \leftarrow \|v\|_2$
4: $\quad v \leftarrow v/\lambda$
5: **end for**
6: **return** $L_f = \|f(v) - f(0)\|_2$

data samples, we can train deep feed-forward networks (with an universal approximation) to fit them, e.g., via the greedy layer-wise training strategy Bengio et al. (2006). Figure 2 visualizes the unrolled deep feed-forward network for the optimization-based data reconstruction attack.

**Definition 1** (Deep Feed-forward Network). *An $H$-layer feed-forward network is an function $T_{MLP}(\mathbf{x}) = f_H \circ \rho_{H-1} \circ \cdots \circ \rho_1 \circ f_1(\mathbf{x})$, where $\forall h$, the $h$-th hidden layer $f_h : \mathbf{x} \mapsto \mathbf{M}_h \mathbf{x} + b_h$ is an affine function and $\rho_h$ is a non-linear activation function.*

**Upper bound Lipschitz computation:** Virmaux & Scaman (2018) showed that computing the exact Lipschitz constant for deep (even 2-layer) feed-forward networks is NP-hard. Hence, they resort to an approximate computation and propose a method called **AutoLip** to obtain an upper bound of the Lipschitz constant. AutoLip relies on the concept of *automatic differentiation* Griewank & Walther (2008), a principled approach that computes differential operators of functions from consecutive operations through a computation graph. When the operations are all locally Lipschitz-continuous and their partial derivatives can be computed and maximized, AutoLip can compute the Lipschitz upper bound efficiently. Algorithm 2 shows the details of AutoLip.

With Autolip, Virmaux & Scaman (2018) showed that a feed-forward network with 1-Lipschitz activation functions has an upper bounded Lipschitz constant below.

**Lemma 1.** *For any $H$-layer feed-forward network with $1$-Lipschitz activation functions, the AutoLip upper bound becomes $L_{AutoLip} = \prod_{h=1}^{H} \|\mathbf{M}_h\|_2$, where $\mathbf{M}_h$ is defined in Definition 1.*

Note that a matrix $\ell_2$-norm equals to its largest singular value, which could be computed efficiently via the *power method* (Mises & Pollaczek-Geiringer, 1929). More details are shown in Algorithm 3[3]. Obviously, the used Clip activation function is 1-Lipschitz. Hence, by applying Lemma 1 to the optimization-based data reconstruction attacks, we can derive an upper bounded Lipschitz $L_{\mathcal{R}}$.

## 5 EVALUATION

### 5.1 EXPERIMENTAL SETUP

**Datasets and models:** We conduct experiments on three benchmark image datasets, i.e., MNIST (Le-Cun, 1998), Fashion-MNIST (FMNIST) (Xiao et al., 2017), and CIFAR10 (Krizhevsky et al., 2009). We examine our theoretical results on the FL algorithm that uses the $\ell_2$-regularized logistic regression ($\ell_2$-LogReg) and the convex 2-layer linear convolutional network (2-LinConvNet) (Pilanci & Ergen, 2020), since their loss functions are convex and satisfy Assumptions 1-4. In the experiments, we evenly distribute the training data among all the $N$ devices. Based on this setting, we can calculate $L, \mu, \sigma_k$, and $G$ in Assumptions 1-4, respectively. In addition, we can compute the Lipschitz constant $L_{\mathcal{R}}$ via the unrolled feed-forward network. These values together are used to compute the upper bound of our Theorems 1 and 2. *More details about the two algorithms, the unrolled feed-forward network, and the calculation of these parameter values are shown in Appendix C.1.*

**Attack baselines:** We test our theoretical results on four optimization-based data reconstruction attacks, i.e., DLG (Zhu et al., 2019), iDLG (Zhao et al., 2020), InvGrad (Geiping et al., 2020), and the GGL (Li et al., 2022). The algorithms and descriptions of these attacks are deferred to Appendix C.1. We test these attacks on recovering both the single image and a batch of images in each device.

---

[3]A better estimation algorithm can lead to a tighter upper bounded Lipschitz constant.

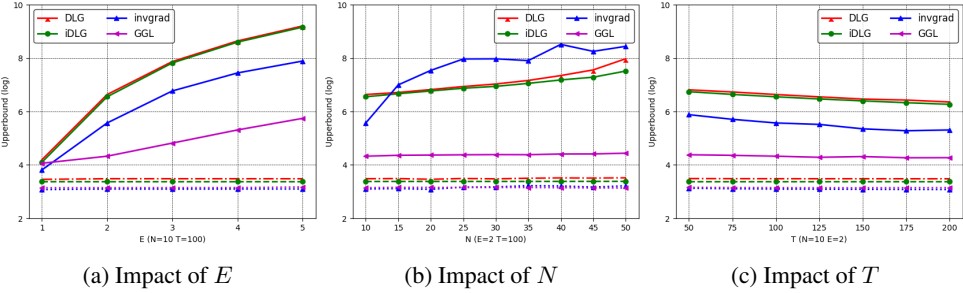

(a) Impact of $E$      (b) Impact of $N$      (c) Impact of $T$

Figure 3: Results of federated $\ell_2$-LogReg on MNIST—single image recovery. Dashed lines are *best* empirical reconstruction errors obtained by existing data reconstruction attacks, while solid lines are *upper bound* errors obtained by our theoretical results. Y-axis is in a log form.

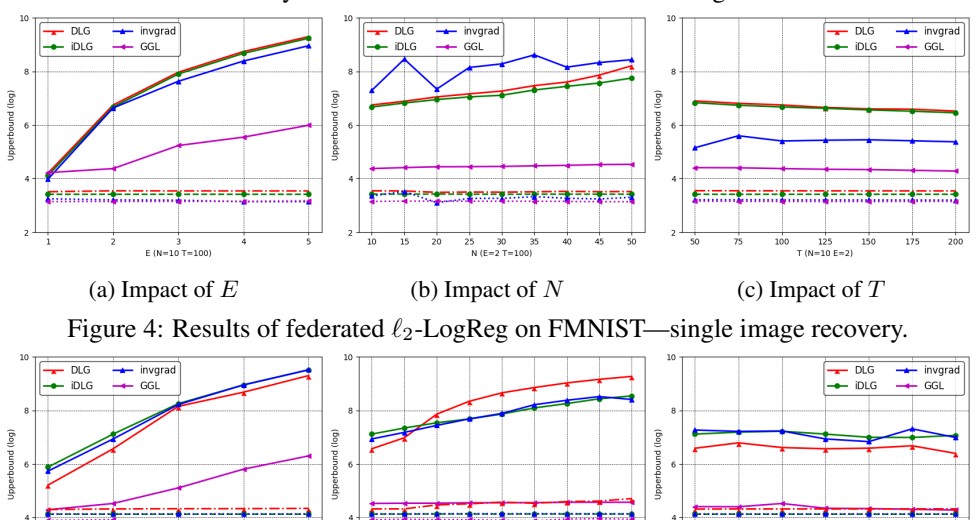

(a) Impact of $E$      (b) Impact of $N$      (c) Impact of $T$

Figure 4: Results of federated $\ell_2$-LogReg on FMNIST—single image recovery.

(a) Impact of $E$      (b) Impact of $N$      (c) Impact of $T$

Figure 5: Results of federated $\ell_2$-LogReg on CIFAR10—single image recovery.

**Parameter setting:** There are several important hyperparameters in the FL training that would affect our theoretical results: the total number of devices $N$, the total number of global rounds $T$, and the number of local SGD updates $E$. By default, we set $T = 100$ and $E = 2$. We set $N = 10$ on the three datasets for the single image recovery, while set $N = 15, 10, 5$ on the three datasets for the batch images recovery, considering their different difficulty levels. We also study the impact of these hyperparameters.

## 5.2 EXPERIMENTAL RESULTS

In this section, we will test the upper bound reconstruction error by our theoretical results. We also show the *best* reconstruction errors that are empirically obtained by the existing well-known attacks. We will show results on both single image recovery and batch images recovery.

### 5.2.1 RESULTS ON THE SINGLE IMAGE RECOVERY

Figures 3-8 show the single image recovery results on the three datasets and two FL algorithms, respectively. We have several observations: *Empirically*, comparing the best reconstruction errors, GGL performs the best; InvGrad is (slightly) smaller than iDLG, which is (slightly) smaller than DLG in most cases. These observations are consistent with those shown in Zhao et al. (2020) and Geiping et al. (2020). This is because GGL uses a pretrained encoder to enforce the reconstructed image to be aligned with natural images. iDLG can first accurately estimate the data label in a closed form and then performs the data reconstruction attack, while DLG needs to estimate both data features and data labels at the same time in an iterative way. Further, on top of iDLG, InvGrad adds a regularization prior to enforce a relatively stable data reconstruction process.

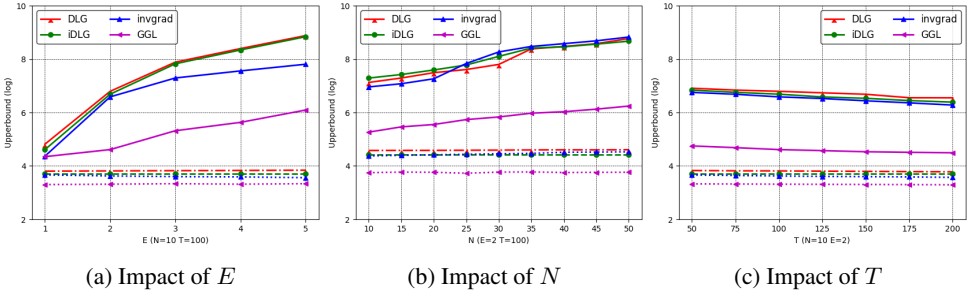

Figure 6: Results of federated 2-LinConvNet on MNIST—single image recovery.

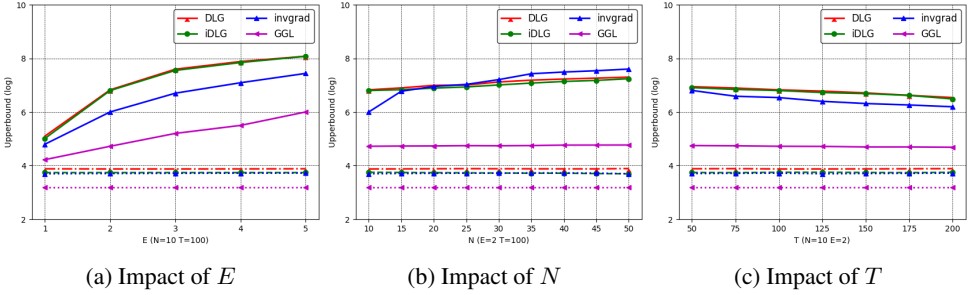

Figure 7: Results of federated 2-LinConvNet on FMNIST—single image recovery.

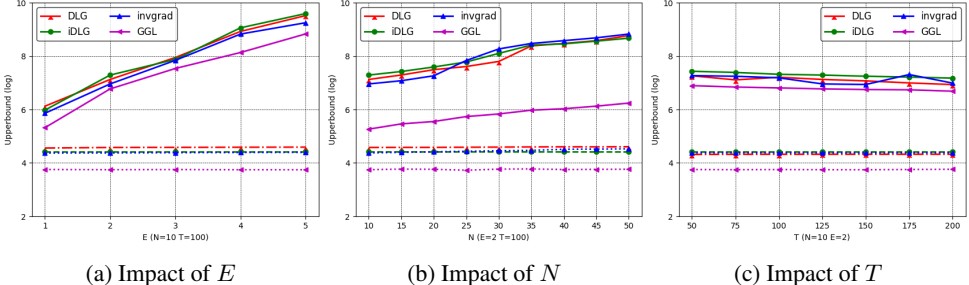

Figure 8: Results of federated 2-LinConvNet on CIFAR10—single image recovery.

*Theoretically*, 1) on one hand, iDLG also has smaller upper bound errors than DLG, indicating iDLG outperforms DLG intrinsically. One possible reason is that iDLG can accurately estimate the labels, which ensures data reconstruction to be more stable. Such a stable reconstruction yields a smaller Lipschitz $L_{\mathcal{R}}$ in Assumption 5, and thus a smaller upper bound in Theorem 1. In contrast, we do not see that InvGrad consistently outperforms iDLG. This implies that enforcing the TV data prior may not be beneficial for theoretical results for single image recovery, as the prior on a single image may not be accurate enough. On the other hand, the error bounds of these three attacks are (much) larger than the empirical ones, indicating that there is still a gap between empirical results and theoretical results. 2) Additionally, GGL has (much) smaller bounded errors than DLG, iDLG, and InvGrad. This is because GGL trains an encoder on the *whole* dataset to learn the image manifold, and then uses the encoder to stabilize the reconstruction, hence producing smaller $L_{\mathcal{R}}$. In certain cases, the bounded error is also close to its best empirical error. 3) The error bounds do not show strong correlations with empirical errors in some cases, e.g., InvGrad on FMNIST in Figure 4. The reason is that the reported empirical errors are the best possible *one-snapshot* results with a certain initialization, which do not reflect the attacks' inherent effectiveness. Recall in Figure 1 that empirical errors obtained by these attacks could be sensitive to different initializations. In practice, the attacker may need to try many initializations (which could be time-consuming) to obtain the best empirical error. However, we show in Figure 9c that the error bounds are consistent with the *average* empirical errors.

**Impact of $E$, $N$, and $T$:** When the local SGD updates $E$ and the number of total clients $N$ increase, the upper bound error also increases. This is because a large $E$ and $N$ will make FL training unstable and hard to converge, as verified in (Li et al., 2020b). On the other hand, a larger total number of global rounds $T$ tends to produce a smaller upper bounded error. This is because a larger $T$ stably makes FL training closer to the global optimal.

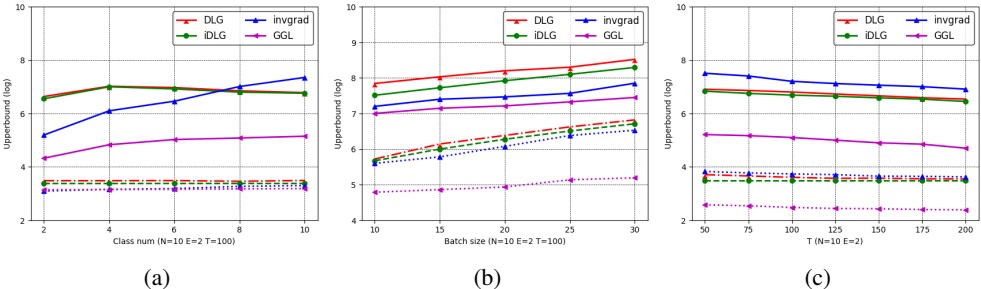

Figure 9: (a) Impact of #classes on MNIST; (b) Impact of batch size on MNIST; (c) Averaged empirical errors vs. bounded errors on FMNIST.

### 5.2.2 RESULTS ON THE BATCH IMAGES RECOVERY

Figure 10-Figure 12 in Appendix C.2 show the results of batch images recovery. Since federated 2-LinConvNet has similar trends, we only show federated $\ell_2$-LogReg results for simplicity. Our key observations are: 1) Similar to results on single image recovery, GGL performs the best both empirically and theoretically; iDLG outperforms DLG both empirically and theoretically, and a larger $E$ and $N$ will incur larger upper bound error, while a larger $T$ will generate smaller upper bound error. 2) Differently, InvGrad theoretically outperforms DLG and iDLG on CIFAR10 for batch images recovery, implying that the data prior enforced by InvGrad is useful in this setting. This is possibly because CIFAR10 is the most complex dataset among the three, and the TV prior could guide the training to be relatively more stable, which hence leads to a smaller $\mathcal{L}_{\mathcal{R}}$. 3) Both the best empirical reconstruction errors and upper bound errors for batch images recovery are much larger than those for single image recovery. This indicates that batch images recovery are more difficult than single image recovery, as validated in many existing works such as Geiping et al. (2020); Yin et al. (2021).

## 6 DISCUSSION

**Error bounds vs. number of classes:** We tested #classes=2, 4, 6, 8 on MNIST and the results are shown in Figure 9a. We can see the bounded errors are relatively stable vs. #classes on DLG, iDLG, and GGL, while InvGrad has a larger error as the #classes increases. The possible reason is that DLG and iDLG are more stable than InvGrad, which involves a more complex optimization.

**Error bounds vs. batch size:** Our batch results use a batch size 20. Here, we also test batch size=10, 15, 25, 30 and results are in Figure 9b. We see bounded errors become larger with larger batch size. This is consistent with existing observations (Geiping et al., 2020) on empirical evaluations.

**Error bounds vs. average empirical error:** As stated, the best one-snapshot empirical errors are not consistent with the bounded errors in some cases for certain attack (e.g., InvGrad on FMNIST). However, we justify that the error bound per attack should have a *strong* correlation with its empirical errors *in expectation*. To verify this, we obtain the expected empirical error per attack by running the attack 10 times and we report the results (in the log form) on FMNIST in Figure 9c. Now, we can see the consistency between the error bounds and average empirical errors.

**Error bounds on closed-form data reconstruction attacks:** Our theoretical results can be also applied in closed-form attacks. Here, we choose the Robbing attack (Fowl et al., 2022) for evaluation and its details are in Appendix C.1.2. The results are shown in Figure 13-Figure 15 in Appendix C.2. We can see Robbing obtains both small empirical errors and bounded errors (which are even smaller than GGL). This is because its equation solving is suitable to linear layers, and hence relatively accurate on the federated $\ell_2$-LogReg and federated 2-LinConvNet models.

## 7 CONCLUSION AND FUTURE WORK

Federated learning (FL) is vulnerable to data reconstruction attacks. Existing attacks mainly enhance the empirical attack performance, but lack a theoretical understanding. We study data reconstruction attacks to FL from a theoretical perspective. Our theoretical results provide a unified way to compare existing attacks theoretically. We also validate our theoretical results via experimental evaluations on multiple image datasets and data reconstruction attacks. Future works include: 1) designing better or adapting existing Lipschitz estimation algorithms to obtain tighter error bounds; 2) generalizing our theoretical results to more challenging settings, e.g., non-convex losses; and 3) designing *theoretically* better data reconstruction attacks (i.e., with smaller Lipschitz) as well as effective defenses against the attacks (i.e., ensuring larger Lipschitz of their reconstruction function), inspired by our framework.

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

---

**Algorithm 4** Optimization-based data reconstruction attacks (e.g., DLG, iDLG, InvGrad, and GGL)

---

**Input:** Model parameters $\mathbf{w}_t$; true gradient $g(\mathbf{x}, y)$; $\eta, \lambda$; public generator $G(\cdot)$, transformation operator $\mathcal{T}$.
**Output:** Reconstructed data $\hat{\mathbf{x}}$.

1: **if DLG then**
2:     $\mathbf{x}'_0 \sim \mathcal{N}(0,1), y'_0 \sim \mathcal{N}(0,1)$
3: **else**
4:     **if GGL then**
5:         $\mathbf{z}'_0 \sim \mathcal{N}(0, \mathbf{I}_u)$;
6:     **else**
7:         $\mathbf{x}'_0 \sim \mathcal{N}(0,1)$ // Initialize dummy input(s)
8:     **end if**
9:     Estimate $y$ as $\hat{y}$ via methods in Zhao et al. (2020) for a single input or Yin et al. (2021) for a batch inputs
10: **end if**
11: **for** $i = 0; i < I; i + +$ **do**
12:     **if DLG then**
13:         $g(\mathbf{x}'_i, y'_i) \leftarrow \nabla_{\mathbf{w}}\mathcal{L}(\mathbf{w}_t; (\mathbf{x}'_i, y'_i))$
14:         $\text{GML}_i \leftarrow \|g(\mathbf{x}, y) - g(\mathbf{x}'_i, y'_i)\|^2_2$
15:         $\mathbf{x}'_{i+1} \leftarrow \mathbf{x}'_i - \eta \nabla_{\mathbf{x}'_i}\text{GML}_i$

        $y'_{i+1} \leftarrow y'_i - \eta \nabla_{y'_i}\text{GML}_i$
16:     **else if iDLG then**
17:         $g(\mathbf{x}'_i, \hat{y}) \leftarrow \nabla_{\mathbf{w}}\mathcal{L}(\mathbf{w}_t; (\mathbf{x}'_i, \hat{y}))$
18:         $\text{GML}_i \leftarrow \|g(\mathbf{x}, y) - g(\mathbf{x}'_i, \hat{y})\|^2_2$
19:         $\mathbf{x}'_{i+1} \leftarrow \mathbf{x}'_i - \eta \nabla_{\mathbf{x}'_i}\text{GML}_i$
20:     **else if InvGrad then**
21:         $g(\mathbf{x}'_i, \hat{y}) \leftarrow \nabla_{\mathbf{w}}\mathcal{L}(\mathbf{w}_t; (\mathbf{x}'_i, \hat{y}))$
22:         $\text{GML}_i \leftarrow 1 - \frac{\langle g(\mathbf{x}, y), g(\mathbf{x}'_i, \hat{y})\rangle}{\|g(\mathbf{x}, y)\|_2 \cdot \|g(\mathbf{x}'_i, \hat{y}\|_2}$
23:         $\mathbf{x}'_{i+1} \leftarrow \mathbf{x}'_i - \eta \nabla_{\mathbf{x}'_i}\left(\text{GML}_i + \lambda \text{Reg}_{\text{TV}}(\mathbf{x}'_i)\right)$
24:     **else if GGL then**
25:         $\mathbf{x}'_i = G(\mathbf{z}'_i)$
26:         $g(\mathbf{x}'_i, \hat{y}) \leftarrow \nabla_{\mathbf{w}}\mathcal{L}(\mathbf{w}_t; (\mathbf{x}'_i, \hat{y}))$
27:         $\text{GML}_i \leftarrow \|g(\mathbf{x}, y) - \mathcal{T}(g(\mathbf{x}'_i, \hat{y}))\|^2_2$
28:         $\text{Reg}(G, \mathbf{z}'_i) = (\|\mathbf{z}_i\|^2_2 - k)^2$
29:         $\mathbf{z}'_{i+1} \leftarrow \mathbf{z}'_i - \eta \nabla_{\mathbf{z}'_i}\left(\text{GML}_i + \lambda \text{Reg}(G, \mathbf{z}'_i)\right)$
30:     **end if**
31:     $\mathbf{x}'_{i+1} = \max(\mathbf{x}'_{i+1}, 0)$
32: **end for**
33: **return** return $\mathbf{x}'_I$ or $G(\mathbf{z}'_I)$

---

## A    PROOF OF THEOREM 1 FOR FULL DEVICE PARTICIPATION

Our proof is mainly inspired by the proofs in Stich (2018); Yu et al. (2019); Li et al. (2020b).

We first restate Theorem 1 as below:

**Theorem 1.** *Let Assumptions 1 to 5 hold. Choose* $\gamma = \max\{8L/\mu, E\}$ *and the learning rate* $\eta_t = \frac{2}{\mu(\gamma+t)}$. *Then, for any communication round $t$, FedAvg with full device participation satisfies*

$$\mathbb{E}\|\mathbf{x} - \mathcal{R}(\mathbf{w}_t)\|^2 \leq 2\mathbb{E}\|\mathbf{x} - \mathcal{R}(\mathbf{w}^\star)\|^2 + \frac{2L_{\mathcal{R}}^2}{\gamma+t}\left(\frac{4B}{\mu^2} + (\gamma+1)\mathbb{E}\|\mathbf{w}_1 - \mathbf{w}^\star\|^2\right), \tag{5}$$

*where* $B = \sum_{k=1}^{N} p_k^2 \sigma_k^2 + 6L\Gamma + 8(E-1)^2 G^2$.

**Notations:** Let $N$ be the total number of user devices and $K(\leq N)$ be the maximal number of devices that participate in every round's communication. Let $T$ be the total number of every device's SGDs, and $E$ be the number of each device's local updates between two communication rounds. Thus $T/E$ is the number of communications, assuming $E$ is dividable by $T$.

Let $\mathbf{w}_t^k$ be the model parameter maintained in the $k$-th device at the $t$-th step. Let $\mathcal{I}_E$ be the set of global aggregation steps, i.e., $\mathcal{I}_E = \{nE \mid n = 1, 2, \cdots\}$. If $t + 1 \in \mathcal{I}_E$, i.e., the devices

communicate with the server and the server performs the `FedAvg` aggregation on device models. Then the update of `FedAvg` with partial devices active can be described as

$$\mathbf{v}_{t+1}^k = \mathbf{w}_t^k - \eta_t \nabla \mathcal{L}_k(\mathbf{w}_t^k, \xi_t^k), \tag{7}$$

$$\mathbf{w}_{t+1}^k = \begin{cases} \mathbf{v}_{t+1}^k & \text{if } t+1 \notin \mathcal{I}_E, \\ \sum_{k=1}^N p_u \mathbf{v}_{t+1}^k & \text{if } t+1 \in \mathcal{I}_E. \end{cases} \tag{8}$$

Motivated by (Stich, 2018; Li et al., 2020b), we define two virtual sequences $\mathbf{v}_t = \sum_{k=1}^N p_k \mathbf{v}_t^k$ and $\mathbf{w}_t = \sum_{k=1}^N p_k \mathbf{w}_t^k$. $\mathbf{v}_{t+1}$ results from an single step of SGD from $\mathbf{w}_t$. When $t+1 \notin \mathcal{I}_E$, both are inaccessible. When $t+1 \in \mathcal{I}_E$, we can only fetch $\mathbf{w}_{t+1}$. For convenience, we define $\overline{\mathbf{g}}_t = \sum_{k=1}^N p_k \nabla \mathcal{L}_k(\mathbf{w}_t^k)$ and $\mathbf{g}_t = \sum_{k=1}^N p_k \nabla \mathcal{L}_k(\mathbf{w}_t^k, \xi_t^k)$. Hence, $\mathbf{v}_{t+1} = \mathbf{w}_t - \eta_t \mathbf{g}_t$ and $\mathbb{E}\mathbf{g}_t = \overline{\mathbf{g}}_t$.

Before proving Theorem 1, we need below key lemmas that are from Stich (2018); Li et al. (2020b).

**Lemma 2** (Results of one step SGD). *Assume Assumptions 1 and 2 hold. If $\eta_t \leq \frac{1}{4L}$, we have*

$$\mathbb{E}\left\|\mathbf{v}_{t+1} - \mathbf{w}^\star\right\|^2 \leq (1 - \eta_t \mu) \mathbb{E}\left\|\mathbf{w}_t - \mathbf{w}^\star\right\|^2 + \eta_t^2 \mathbb{E}\left\|\mathbf{g}_t - \overline{\mathbf{g}}_t\right\|^2 + 6L\eta_t^2 \Gamma + 2\mathbb{E}\sum_{k=1}^N p_k \left\|\mathbf{w}_t - \mathbf{w}_k^t\right\|^2$$

*where $\Gamma = \mathcal{L}^* - \sum_{k=1}^N p_k \mathcal{L}_k^\star \geq 0$.*

*Proof sketch:* Lemma 2 is mainly from Lemma 1 in Li et al. (2020b). The proof idea is to bound three terms, i.e., the inner product $\langle \mathbf{w}_t - \mathbf{w}^*, \nabla \mathcal{L}(\mathbf{w}_t) \rangle$, the square norm $||\nabla \mathcal{L}(\mathbf{w}_t)||^2$, and the inner product $\langle \nabla \mathcal{L}_k(\mathbf{w}_t), \mathbf{w}_t^k - \mathbf{w}_t \rangle, \forall k$. Then, the left-hand term in Lemma 2 can be rewritten in terms of the three terms and be bounded by the right-hand four terms in Lemma 2. Specifically, 1) It first bounds $\langle \mathbf{w}_t - \mathbf{w}^*, \nabla \mathcal{L}(\mathbf{w}_t) \rangle$ using the strong convexity of the loss function (Assumption 2); 2) It bounds $||\nabla \mathcal{L}(\mathbf{w}_t)||^2$ using the smoothness of the loss function (Assumption 1); and 3) It bounds $\langle \nabla \mathcal{L}_k(\mathbf{w}_t), \mathbf{w}_t^k - \mathbf{w}_t \rangle, \forall k$ using the convexity of the loss function (Assumption 2).

**Lemma 3** (Bounding the variance). *Assume Assumption 3 holds. Then $\mathbb{E}\left\|\mathbf{g}_t - \overline{\mathbf{g}}_t\right\|^2 \leq \sum_{k=1}^N p_u^2 \sigma_u^2$.*

**Lemma 4** (Bounding the divergence of $\{\mathbf{w}_t^k\}$). *Assume Assumption 4 holds, and $\eta_t$ is non-increasing and $\eta_t \leq 2\eta_{t+E}$ for all $t \geq 0$. It follows that $\mathbb{E}\left[\sum_{k=1}^N p_k \left\|\mathbf{w}_t - \mathbf{w}_k^t\right\|^2\right] \leq 4\eta_t^2(E-1)^2 G^2$.*

Now, we complete the proof of Theorem 1.

*Proof.* First, we observe that no matter whether $t+1 \in \mathcal{I}_E$ or $t+1 \notin \mathcal{I}_E$ in Equation (8), we have $\mathbf{w}_{t+1} = \mathbf{v}_{t+1}$. Denote $\Delta_t = \mathbb{E}\left\|\mathbf{w}_t - \mathbf{w}^\star\right\|^2$. From Lemmas 2 to 4, we have

$$\Delta_{t+1} = \mathbb{E}\left\|\mathbf{w}_{t+1} - \mathbf{w}^\star\right\|^2 = \mathbb{E}\left\|\mathbf{v}_{t+1} - \mathbf{w}^\star\right\|^2 \leq (1 - \eta_t \mu)\Delta_t + \eta_t^2 B \tag{9}$$

where $B = \sum_{k=1}^N p_u^2 \sigma_u^2 + 6L\Gamma + 8(E-1)^2 G^2$.

For a diminishing stepsize, $\eta_t = \frac{\beta}{t+\gamma}$ for some $\beta > \frac{1}{\mu}$ and $\gamma > 0$ such that $\eta_1 \leq \min\{\frac{1}{\mu}, \frac{1}{4L}\} = \frac{1}{4L}$ and $\eta_t \leq 2\eta_{t+E}$. We will prove $\Delta_t \leq \frac{v}{\gamma + t}$ where $v = \max\left\{\frac{\beta^2 B}{\beta\mu - 1}, (\gamma+1)\Delta_1\right\}$.

We prove it by induction. Firstly, the definition of $v$ ensures that it holds for $t = 1$. Assume the conclusion holds for some $t$, it follows that

$$\Delta_{t+1} \leq (1 - \eta_t \mu)\Delta_t + \eta_t^2 B$$

$$\leq \left(1 - \frac{\beta\mu}{t+\gamma}\right)\frac{v}{t+\gamma} + \frac{\beta^2 B}{(t+\gamma)^2}$$

$$= \frac{t+\gamma-1}{(t+\gamma)^2}v + \left[\frac{\beta^2 B}{(t+\gamma)^2} - \frac{\beta\mu-1}{(t+\gamma)^2}v\right]$$

$$\leq \frac{v}{t+\gamma+1}.$$

By the $\bar{L}$-Lipschitz continuous property of $\text{Rec}(\cdot)$,

$$\|\text{Rec}(\mathbf{w}_t) - \text{Rec}(\mathbf{w}^*)\| \leq \bar{L} \cdot \|\mathbf{w}_t - \mathbf{w}^\star\|.$$

Then we have

$$\mathbb{E}\|\text{Rec}(\mathbf{w}_t) - \text{Rec}(\mathbf{w}^*)\|^2 \le \bar{L}^2 \cdot \mathbb{E}\|\mathbf{w}_t - \mathbf{w}^\star\|^2 \le \bar{L}^2 \Delta_t \le \bar{L}^2 \frac{v}{\gamma + t}.$$

Specifically, if we choose $\beta = \frac{2}{\mu}, \gamma = \max\{8\frac{L}{\mu}, E\} - 1$, then $\eta_t = \frac{2}{\mu}\frac{1}{\gamma+t}$. We also verify that the choice of $\eta_t$ satisfies $\eta_t \le 2\eta_{t+E}$ for $t \ge 1$. Then, we have

$$v = \max\left\{\frac{\beta^2 B}{\beta\mu - 1}, (\gamma + 1)\Delta_1\right\} \le \frac{\beta^2 B}{\beta\mu - 1} + (\gamma + 1)\Delta_1 \le \frac{4B}{\mu^2} + (\gamma + 1)\Delta_1.$$

Hence,

$$\mathbb{E}\|\text{Rec}(\mathbf{w}_t) - \text{Rec}(\mathbf{w}^*)\|^2 \le \bar{L}^2 \frac{v}{\gamma + t} \le \frac{\bar{L}^2}{\gamma + t}\left(\frac{4B}{\mu^2} + (\gamma + 1)\Delta_1\right).$$

$\square$

## B  PROOFS OF THEOREM 2 FOR PARTIAL DEVICE PARTICIPATION

We first restate Theorem 2 as below:

**Theorem 2.** *Let Assumptions 1 to 6 hold. Let $\gamma$, $\eta_t$, and $B$ be defined in Theorem 1, and define $C = \frac{4}{K}E^2G^2$. Then, for any communication round $t$,*

$$\mathbb{E}\|\mathbf{x} - \mathcal{R}(\mathbf{w}_t)\|^2 \le 2\mathbb{E}\|\mathbf{x} - \mathcal{R}(\mathbf{w}^\star)\|^2 + \frac{2L_\mathcal{R}^2}{\gamma + t}\left(\frac{4(B + C)}{\mu^2} + (\gamma + 1)\mathbb{E}\|\mathbf{w}_1 - \mathbf{w}^*\|^2\right). \quad (6)$$

Recall that $\mathbf{w}_t^k$ is $k$-th device's model at the $t$-th step, $\mathcal{I}_E = \{nE \mid n = 1, 2, \cdots\}$ is the set of global aggregation steps; $\overline{\mathbf{g}}_t = \sum_{k=1}^N p_k \nabla \mathcal{L}_k(\mathbf{w}_t^k)$ and $\mathbf{g}_t = \sum_{k=1}^N p_k \mathcal{L}_k(\mathbf{w}_t^k, \xi_t^k)$, and $\mathbf{v}_{t+1} = \mathbf{w}_t - \eta_t \mathbf{g}_t$ and $\mathbb{E}\mathbf{g}_t = \overline{\mathbf{g}}_t$. We denote by $\mathcal{H}_t$ the multiset selected which allows any element to appear more than once. Note that $\mathcal{H}_t$ is only well defined for $t \in \mathcal{I}_E$. For convenience, we denote by $\mathcal{S}_t = \mathcal{H}_{N(t,E)}$ the most recent set of chosen devices where $N(t, E) = \max\{n|n \le t, n \in \mathcal{I}_E\}$.

In partial device participation, FedAvg first samples a random multiset $\mathcal{S}_t$ of devices and then only performs updates on them. Directly analyzing on the $\mathcal{S}_t$ is complicated. Motivated by Li et al. (2020b), we can use a thought trick to circumvent this difficulty. Specifically, we assume that FedAvg always activates *all devices* at the beginning of each round and uses the models maintained in only a few sampled devices to produce the next-round model. It is clear that this updating scheme is equivalent to that in the partial device participation. Then the update of FedAvg with partial devices activated can be described as:

$$\mathbf{v}_{t+1}^k = \mathbf{w}_t^k - \eta_t \nabla \mathcal{L}_k(\mathbf{w}_t^k, \xi_t^k), \quad (10)$$

$$\mathbf{w}_{t+1}^k = \left\{\begin{array}{ll} \mathbf{v}_{t+1}^k & \text{if } t + 1 \notin \mathcal{I}_E, \\ \text{samples } \mathcal{S}_{t+1} \text{ and average } \{\mathbf{v}_{t+1}^k\}_{k \in \mathcal{S}_{t+1}} & \text{if } t + 1 \in \mathcal{I}_E. \end{array}\right. \quad (11)$$

Note that in this case, there are two sources of randomness: stochastic gradient and random sampling of devices. The analysis for Theorem 1 in Appendix A only involves the former. To distinguish with it, we use an extra notation $\mathbb{E}_{\mathcal{S}_t}(\cdot)$ to consider the latter randomness.

First, based on Li et al. (2020b), we have the following two lemmas on unbiasedness and bounded variance. Specifically, Lemma 5 shows the scheme in Assumption 6 is unbiased, while Lemma 6 shows the expected difference between $\mathbf{v}_{t+1}$ and $\mathbf{w}_{t+1}$ is bounded.

**Lemma 5** (Unbiased sampling scheme in Assumption 6). *If $t+1 \in \mathcal{I}_E$, we have $\mathbb{E}_{\mathcal{S}_t}(\mathbf{w}_{t+1}) = \mathbf{v}_{t+1}$.*

**Lemma 6** (Bounding the variance of $\mathbf{w}_t$). *For $t + 1 \in \mathcal{I}$, assume that $\eta_t$ is non-increasing and $\eta_t \le 2\eta_{t+E}$ for all $t \ge 0$. Then the expected difference between $\mathbf{v}_{t+1}$ and $\mathbf{w}_{t+1}$ is bounded by $\mathbb{E}_{\mathcal{S}_t}\|\mathbf{v}_{t+1} - \mathbf{w}_{t+1}\|^2 \le \frac{4}{K}\eta_t^2 E^2 G^2$.*

Now, we complete the proof of Theorem 2.

*Proof.* Note that

$$\|\mathbf{w}_{t+1} - \mathbf{w}^*\|^2 = \|\mathbf{w}_{t+1} - \mathbf{v}_{t+1} + \mathbf{v}_{t+1} - \mathbf{w}^*\|^2$$
$$= \underbrace{\|\mathbf{w}_{t+1} - \mathbf{v}_{t+1}\|^2}_{A_1} + \underbrace{\|\mathbf{v}_{t+1} - \mathbf{w}^*\|^2}_{A_2} + \underbrace{2\langle\mathbf{w}_{t+1} - \mathbf{v}_{t+1}, \mathbf{v}_{t+1} - \mathbf{w}^*\rangle}_{A_3}.$$

When expectation is taken over $\mathcal{S}_{t+1}$, the last term ($A_3$) vanishes due to the unbiasedness of $\mathbf{w}_{t+1}$.

If $t+1 \notin \mathcal{I}_E$, $A_1$ vanishes since $\mathbf{w}_{t+1} = \mathbf{v}_{t+1}$. We use Lemma 6 to bound $A_2$. Then it follows that

$$\mathbb{E}\|\mathbf{w}_{t+1} - \mathbf{w}^*\|^2 \leq (1 - \eta_t\mu)\mathbb{E}\|\mathbf{w}_t - \mathbf{w}^\star\|^2 + \eta_t^2 B.$$

If $t+1 \in \mathcal{I}_E$, we additionally use Lemma 6 to bound $A_1$. Then

$$\mathbb{E}\|\mathbf{w}_{t+1} - \mathbf{w}^*\|^2 = \mathbb{E}\|\mathbf{w}_{t+1} - \mathbf{v}_{t+1}\|^2 + \mathbb{E}\|\mathbf{v}_{t+1} - \mathbf{w}^*\|^2$$
$$\leq (1 - \eta_t\mu)\mathbb{E}\|\mathbf{w}_t - \mathbf{w}^\star\|^2 + \eta_t^2 B + \frac{4}{K}\eta_t^2 E^2 G^2$$
$$= (1 - \eta_t\mu)\mathbb{E}\|\mathbf{w}_t - \mathbf{w}^\star\|^2 + \eta_t^2(B + C), \tag{12}$$

where $C = \frac{4}{K}E^2 G^2$ is the upper bound of $\frac{1}{\eta_t^2}\mathbb{E}_{\mathcal{S}_t}\|\mathbf{v}_{t+1} - \mathbf{w}_{t+1}\|^2$.

We observe that the only difference between eqn. (12) and eqn. (9) is the additional $C$. Thus we can use the same argument there to prove the theorems here. Specifically, for a diminishing stepsize, $\eta_t = \frac{\beta}{t+\gamma}$ for some $\beta > \frac{1}{\mu}$ and $\gamma > 0$ such that $\eta_1 \leq \min\{\frac{1}{\mu}, \frac{1}{4L}\} = \frac{1}{4L}$ and $\eta_t \leq 2\eta_{t+E}$, we can prove $\mathbb{E}\|\mathbf{w}_{t+1} - \mathbf{w}^*\|^2 \leq \frac{v}{\gamma+t}$ where $v = \max\left\{\frac{\beta^2(B+C)}{\beta\mu-1}, (\gamma+1)\|\mathbf{w}_1 - \mathbf{w}^*\|^2\right\}$.

Then by the $\bar{L}$-Lipschitz continuous property of $\text{Rec}(\cdot)$,

$$\mathbb{E}\|\text{Rec}(\mathbf{w}_t) - \text{Rec}(\mathbf{w}^*)\|^2 \leq \bar{L}^2 \cdot \mathbb{E}\|\mathbf{w}_t - \mathbf{w}^\star\|^2 \leq \bar{L}^2\Delta_t \leq \bar{L}^2\frac{v}{\gamma+t}.$$

Specifically, if we choose $\beta = \frac{2}{\mu}, \gamma = \max\{8\frac{L}{\mu}, E\} - 1$,

$$\mathbb{E}\|\text{Rec}(\mathbf{w}_t) - \text{Rec}(\mathbf{w}^*)\|^2 \leq \bar{L}^2\frac{v}{\gamma+t} \leq \frac{\bar{L}^2}{\gamma+t}\left(\frac{4(B+C)}{\mu^2} + (\gamma+1)\|\mathbf{w}_1 - \mathbf{w}^*\|^2\right).$$

$\square$

## C EXPERIMENTS

### C.1 MORE EXPERIMENTAL SETUP

#### C.1.1 DETAILS ABOUT THE FL ALGORITHMS AND UNROLLED FEED-FORWARD NETWORK

We first show how to compute calculate $L, \mu, \sigma_u$, and $G$ in Assumptions 1-4 on federated $\ell_2$-regularized logistic regression ($\ell_2$-LogReg) and federated 2-layer linear convolutional network (2-LinConvNet); Then we show how to compute the Lipschitz $L_\mathcal{R}$ on each data reconstruction attack.

**Federated $\ell_2$-LogReg:** Each device $k$'s local objective is $\mathcal{L}_k(\mathbf{w}) = \frac{1}{\bar{n}}\sum_{j=1}^{\bar{n}}\log(1 + \exp(-y_j\langle\mathbf{w}, \mathbf{x}_j^k\rangle)) + \gamma\|\mathbf{w}\|^2$. In our results, we simply set $\gamma = 0.1$ for brevity.

- **Compute $L$:** first, all $\mathcal{L}_k$'s are $\frac{1}{4}(\frac{1}{\bar{n}}\sum_j\|x_j^k\|^2)$-smooth (Papailiopoulos, 2018); then $L = \max_{k\in[N]}\frac{1}{4}(\frac{1}{\bar{n}}\sum_j\|\mathbf{x}_j^k\|^2) + 2\gamma$;
- **Compute $\mu$:** all $\mathcal{L}_k$'s are $\gamma$-strongly convex for the $\gamma$ regularized $\ell_2$ logistic regression (Papailiopoulos, 2018) and $\mu = \gamma$.
- **Compute $\sigma^k$ and $G$:** we first traverse all training data $\xi_t^k$ in the $k$-th device in any $t$-th round and then use them to calculate the maximum square norm differences $\|\nabla\mathcal{L}_k(w_t^k, \xi_t^k) - \nabla\mathcal{L}_k(w_t^k)\|^2$. Similarly, $G$ can be calculated as the maximum value of the expected square norm $\|\nabla\mathcal{L}_k(w_t^k, \xi_t^k)\|^2$ among all devices $\{k\}$ and rounds $\{t\}$.

**Federated 2-LinConvNet (Pilanci & Ergen, 2020)**. Let a two-layer network $f : \mathbb{R}^d \to \mathbb{R}$ with $m$ neurons be: $f(\mathbf{x}) = \sum_{j=1}^m \phi(\mathbf{x}^T \mathbf{u}_j) \alpha_j$, where $\mathbf{u}_j \in \mathbb{R}^d$ and $\alpha_j \in \mathbb{R}$ are the weights for hidden and output layers, and $\phi(\cdot)$ is an activation function. Two-layer convolutional networks with $U$ filters can be described by patch matrices (e.g., images) $\mathbf{X}_u, u = 1, \cdots, U$. For flattened activations, we have $f(\mathbf{X}_1, \cdots \mathbf{X}_u) = \sum_{u=1}^U \sum_{j=1}^m \phi(\mathbf{X}_u \mathbf{u}_j) \alpha_j$.

We consider the 2-layer linear convolutional networks and its non-convex loss is defined as:

$$\min_{\{\alpha_j, \mathbf{u}_j\}_{j=1}^m} \mathcal{L}(\{\alpha_j, \mathbf{u}_j\}) = \frac{1}{2} \| \sum_{u=1}^U \sum_{j=1}^m \mathbf{X}_u \mathbf{u}_j \alpha_{ju} - \mathbf{y} \|_2^2. \tag{13}$$

Pilanci & Ergen (2020) show that the above non-convex problem can be transferred to the below convex optimization problem via its duality. and the two problems have the identical optimal values:

$$\min_{\{\mathbf{w}_u \in \mathbb{R}^d\}_{u=1}^U} \mathcal{L}(\{\mathbf{w}_u\}) = \frac{1}{2} \| \sum_{u=1}^U \mathbf{X}_u \mathbf{w}_u - \mathbf{y} \|_2^2. \tag{14}$$

We run federated learning with convex 2-layer linear convolutional network, where each device trains the local loss $\mathcal{L}_k(\{\mathbf{w}_u\}_{u=1}^U)$ and it can converge to the optimal model $\mathbf{w}^* = \{\mathbf{w}_u^*\}$.

- **Compute $L$**: Let $\underline{\mathbf{w}} = \{\mathbf{w}_u\}_{u=1}^U$. For each client $k$, we require its local loss $\mathcal{L}_k$ should satisfy $\|\triangledown \mathcal{L}_k(\underline{\mathbf{w}}) - \triangledown \mathcal{L}_k(\underline{\mathbf{v}})\|_2 \leq L_k \|\underline{\mathbf{w}} - \underline{\mathbf{v}}\|_2$ for any $\underline{\mathbf{w}}, \underline{\mathbf{v}}$; With Equation 14, we have $\left\| \sum_{u=1}^U (\mathbf{X}_u^k)^T \mathbf{X}_u^k (\underline{\mathbf{w}} - \underline{\mathbf{v}}) \right\|_2 \leq L_k \|\underline{\mathbf{w}} - \underline{\mathbf{v}}\|_2$; Then we have the smoothness constant $L_k$ to be the maximum eigenvalue of $\sum_{u=1}^U (\mathbf{X}_u^k)^T \mathbf{X}_u^k$, which is $\| \sum_{u=1}^U (\mathbf{X}_u^k)^T \mathbf{X}_u^k \|_2$; Hence, $L = \max_k \| \sum_{u=1}^U (\mathbf{X}_u^k)^T \mathbf{X}_u^k \|_2$.

- **Compute $\mu$**: Similar as computing $L$, $\mu$ is the minimum eigenvalue of $\sum_{u=1}^U (\mathbf{X}_u^k)^T \mathbf{X}_u^k$ for all $k$, that is, $\mu = \min_k \| \sum_{u=1}^U (\mathbf{X}_u^k)^T \mathbf{X}_u^k \|_2$.

- **Compute $\sigma^k$ and $G$**: Similar computation as in $\ell_2$-regularized logistic regression.

**Unrolled feed-forward network and its training and performance.** In our experiments, we set the number of layers to be 20 in the unrolled feed-forward network for the three datasets. We use 1,000 data samples and their intermediate reconstructions to train the network. To reduce overfitting, we use the greedy layer-wise training strategy. For instance, the average MSE (between the input and output of the unrolled network) of DLG, iDLG, InvGrad, and GGL on MNIST is 1.22, 1.01, 0.76, and 0.04, respectively—indicating that the training performance is promising. After training the unrolled network, we use the AutoLip algorithm to calculate the Lipschitz $L_\mathcal{R}$.

### C.1.2 DETAILS ABOUT DATA RECONSTRUCTION ATTACKS

**GGL (Li et al., 2022)**: GGL considers the scenario where clients realize the server will infer their private data and they hence perturb their local models before sharing them with the server as a defense. To handle noisy models, GGL solves an optimization problem similar to InvGrad, but uses a pretrained generator as a regularization. The generator is trained on the entire MNIST dataset and can calibrate the reconstructed noisy image to be within the image manifold. Specifically, given a well-trained generator $G(\cdot)$ on public datasets and assume the label $y$ is inferred by iGLD, GGL targets the following optimization problem:

$$\mathbf{z}^* = \arg \min_{\mathbf{z} \in \mathbb{R}^k} \text{GML}(g(\mathbf{x}, y), \mathcal{T}(g(G(\mathbf{z}), y))) + \lambda \text{Reg}(G; \mathbf{z}), \tag{15}$$

where $\mathbf{z}$ is the latent space of the generative model, $\mathcal{T}$ is a lossy transformation (e.g., compression or sparsification) acting as a defense, and $\text{Reg}(G; \mathbf{z})$ is a regularization term that penalizes the latent $\mathbf{z}$ if it deviates from the prior distribution. Once the optimal $\mathbf{z}^*$ is obtained, the image can be reconstructed as $G(\mathbf{z}^*)$ and should well align the natural image.

In the experiments, we use a public pretrained GAN generator for MNIST, Fashion-MNIST, and CIFAR. We adopt gradient clipping as the defense strategy $\mathcal{T}$ performed by the clients. Specifically, $\mathcal{T}(g, S) = g / \max(1, \|g\|_2 / S)$. Note that since $G(\cdot)$ is trained on the whole image dataset, it produces stable reconstruction during the optimization.

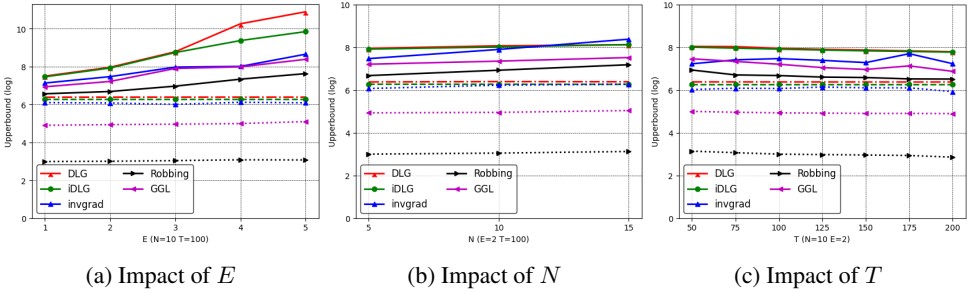

(a) Impact of $E$        (b) Impact of $N$        (c) Impact of $T$

Figure 10: Results of federated $\ell_2$-LogReg on MNIST—batch images recovery. Dashed lines are *best* empirical reconstruction errors obtained by existing data reconstruction attacks, while solid lines are *upper bound* errors obtained by our theoretical results. Y-axis is in a log form. Similarly, we observe that GGL performs the best both empirically and theoretically, due to its pretrained encoder uses the whole dataset to force a stable reconstruction. iDLG (slightly) outperforms DLG both empirically and theoretically; a larger $E$ and $N$ will incur larger upper bound error, while a larger $T$ will generate smaller upper bound error. Additionally, InvGrad theoretically outperforms DLG and iDLG on CIFAR10, indicating that the data prior on complex datasets could be useful.

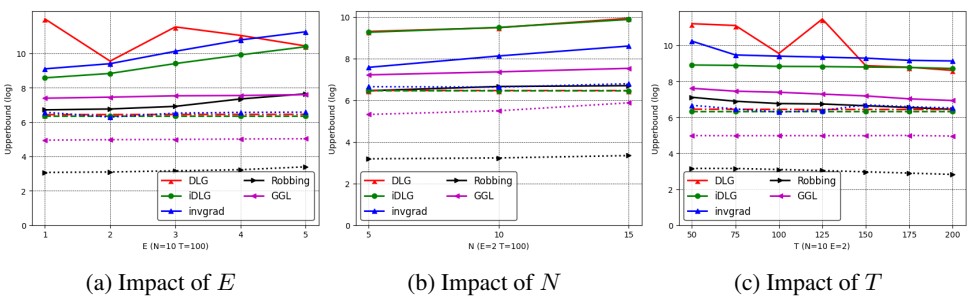

(a) Impact of $E$        (b) Impact of $N$        (c) Impact of $T$

Figure 11: Results of federated $\ell_2$-LogReg on FMNIST—batch images recovery.

**Robbing (Fowl et al., 2022):** Robbing approximately reconstructs the data via solving an equation without any iterative optimization. Assume a batch of data $\mathbf{x}_1, \mathbf{x}_2, \cdots \mathbf{x}_n$ with unique labels $\mathbf{y}_1, \mathbf{y}_2, \cdots \mathbf{y}_n$ in the form of one-hot encoding. Let $\oslash$ be element-wise division. Then, Robbing observes that each row $i$ in $\frac{\partial \mathcal{L}_t}{\partial y_t}$, i.e., $\frac{\partial \mathcal{L}_t}{\partial y_t^i}$, actually recovers

$$\mathbf{x}_t = \frac{\partial \mathcal{L}_t}{\partial y_t^i} \mathbf{x}_t \oslash \frac{\partial \mathcal{L}_t}{\partial y_t^i}.$$

In other others, Robbing directly maps the model to the reconstructed data. Hence, in our experiment, the unrolled feed-forward neural network reduces to 1-layer ReLU network. We then estimate Lipschitz upper bound on this network.

## C.2 MORE EXPERIMENTAL RESULTS

Figure 10 to Figure 12 show the batch images recovery results by the four considered data reconstruction attacks on federated $\ell_2$-LogReg.

Figure 13 to Figure 15 show the recovery results by Robbing on federated $\ell_2$-LogReg and federated 2-LinConvNet.

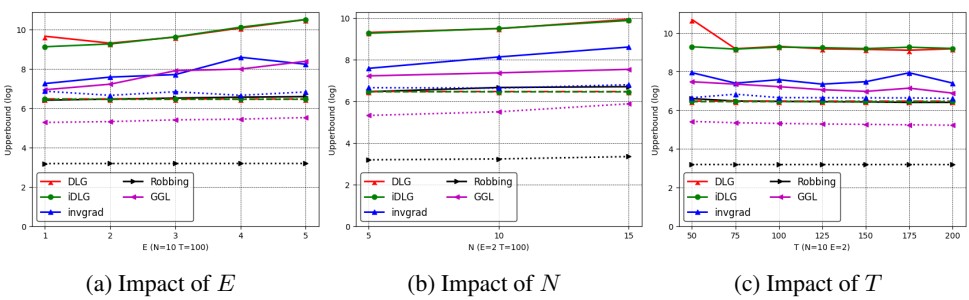

(a) Impact of $E$      (b) Impact of $N$      (c) Impact of $T$

Figure 12: Results of federated $\ell_2$-LogReg on CIFAR10—batch images recovery.

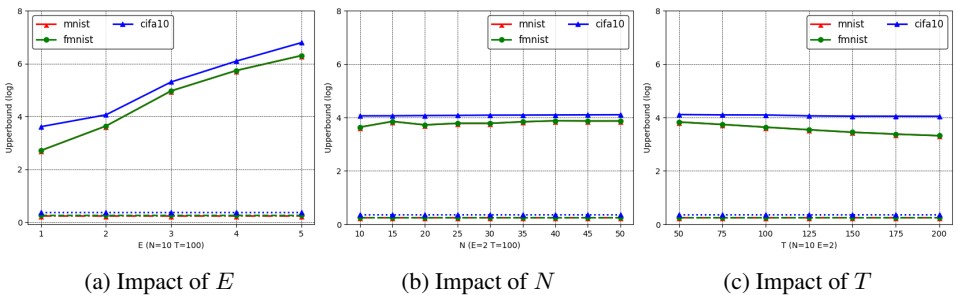

(a) Impact of $E$      (b) Impact of $N$      (c) Impact of $T$

Figure 13: Results of federated $\ell_2$-LogReg on Robbing—single image recovery. We observe that Robbing has much smaller bounded errors and is even smaller than GGL (See Figure 3-Figure 5). This is because the equation solving used by Robbing is accurate on the simple federated $\ell_2$-LogReg model that uses a linear layer.

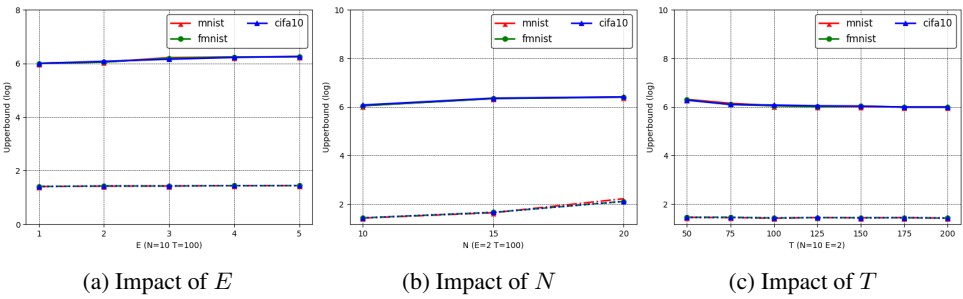

(a) Impact of $E$      (b) Impact of $N$      (c) Impact of $T$

Figure 14: Results of federated $\ell_2$-LogReg on Robbing—batch images recovery.

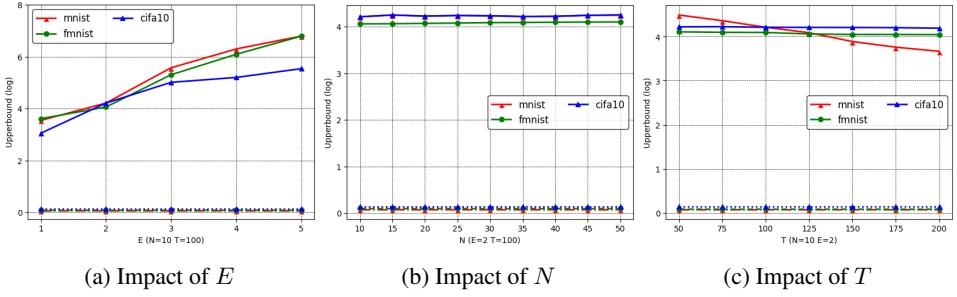

(a) Impact of $E$      (b) Impact of $N$      (c) Impact of $T$

Figure 15: Results of federated 2-LinConvNet on Robbing—single image recovery. Robbing has much smaller error bounds on federated 2-LinConvNet due to its accurate equation solving.

