# OpenReview forum: "Theoretically Understanding Data Reconstruction Leakage in Federated Learning"
_ICLR.cc/2024/Conference — Submitted to ICLR 2024_

### Official Review · Reviewer_sB6V · 2023-10-21

**Soundness:** 2 fair
**Presentation:** 3 good
**Contribution:** 3 good
**Rating:** 5
**Confidence:** 5

**Summary:**

The paper establishes a bound on the success of different gradient leakage attacks at different communication rounds $t$ and proposes it as a measurement of the severity of those attacks. The bound proposed is based on two components - the attack performance at the optimal weights and a term related to the properties of convergence of the underline FedAvg algorithm. The bound depends on several properties of the model (e.g. smoothness and convexity), its gradients (e.g variance and maximum size) and the lipschitzness of the attack function. The authors show experiments with VERY basic models where the properties are true and their parameters can be calculated. For those models the authors compare their bounds to the empirical performance on many gradient leakage attacks and show that certain patterns are shared between the two models.

**Strengths:**

- The authors identify an important problem in federated learning and propose a novel solution
- The authors directly bound the reconstruction error of different gradient leakage algorithms
- The authors demonstrate their bound is applicable on a large class of leakage attacks, even on some analytical ones.

**Weaknesses:**

- **Dependence on the performance at $w^\*$**:
The first part of the bound ($\mathbb{E}[x-\mathcal{R}(w^\*)]$) still depends on the performance of the leakage algorithm $\mathcal{R}$. This means that, in practice, the computational complexity of obtaining the proposed bound is the same as the complexity of the traditional evaluation of leakage attacks (which is very high, especially for attacks based on optimization). It also means that the issues related to estimating the performance of $\mathcal{R}$ in the presence of randomness ( e.g. from the initialization or the choice of the client data batch) remain as hard to solve for the proposed bound as for the original leakage problem. Further, the problem of choosing the optimal hyperparameters for leakage attacks in the presence of such randomness, which would have been one of the best applications of the proposed bound, also remains as hard as before. Finally, the dependency also introduces possible additional challenges compared to evaluating the attack directly on $w_t$ - in particular, estimating $w^\*$. While for convex models, such as the ones explored in the paper, estimating $w^\*$ is not hard, for models with multiple local minima and equivalent solutions ( like generic neural networks ), it is actually challenging to estimate $w^\*$, and it represents an additional source of randomness.
- **The bound's dependence on $t$:**
In Section 5 of the paper, the authors demonstrate, both practically and theoretically, that their bound predicts that as $t$ increases, the vulnerability of the attacked model increases. This is in stark contradiction with the empirical observations about gradient leakage attacks where exactly the opposite is true (e.g. See [1,2]). This mismatch is caused by the second part of the bound, which is supposed to precisely capture the dependence of the attack's success through time but, in reality, is based solely on the convergence of FedAvg and disregards any knowledge of $\mathcal{R}$ but its Lipschitzness. As such, the bound does not capture the evolution of $\mathcal{R}$ with time, only the evolution of the weights.
- **Unclear or missing implementation details:**
1. When the first term of the bound $\mathbb{E}[x-\mathcal{R}(w^\*)]$ is estimated, what is the average taken over? Multiple batches of the client data? Multiple initializations of the algorithms? Across different attack hyperparameters? All?
2. How is $\Gamma$ approximated in practice? How is heterogeneity controlled for in general in the experiments provided? Can you demonstrate the results of experiments on different levels of heterogeneity?
3. The authors in Figure 2 and Algorithm 1 talk about their unrolled network to have parameters $\theta^i$. Where are those parameters coming from, and what do they represent for the used leakage attacks? Even in Line 4 of Algorithm 1, where they are defined, they don't seem to be used. Also, can you elaborate on why you tune them with layer-wise methods instead of SGD? The authors just say "better generalizability" with no context.

- **Problems with the presented evaluation:**
1. All empirical results are presented in the plots on a scale, which makes it very hard to interpret them. In particular, I suggest that the authors use two different scales for the bounds and the empirical results. They can still present both results in the same plot for comparison reasons, but they can show the empirical scale on the left part of the figure and the bound scale on the right. This will enable better comparison between the trends in the two modes of evaluation, as now the empirical models always look completely flat.
2. The paper's main claim is that the proposed bound is a useful tool for evaluating the practical performance of gradient leakage attacks. Yet, the authors do not provide a correlation metric between their bound and the empirical evaluation results. Can the authors precisely measure how correlated their bound is to the actual gradient leakage results?
3. While the paper focuses on analyzing how gradient leakage performance changes with $E$, $N$, and $t$, I want to see their bounds used for comparing the same gradient leakage attack on different models and architectures, as well as, across different hyperparameters such as initialization strategies, regularizer strengths, etc. See [3].
4. The authors claim that one needs to interpret their bound as **average** and not **best-case** reconstruction performance ( Section 6 in the paper ), yet they only provide a single experiment ( Figure 9c ) where they compare against average reconstruction performance. All experiments in the paper should show average behavior for the author's claims to be substantiated.
- **Poor experimental results:**
The current experimental results are weak. What I mean by this is that in many experiments, the bounds do not well reflect what happens to practical performance. For example, in Figures 3 and 4, the empirical evaluation puts invGrad and GGL very close to each other in terms of performance, with invGrad sometimes even better. At the same time, the bounds consistently put the performance of invGrad to be similar to DLG and iDLG. Similarly, in Figures 5a and 5c, the bound predicts better reconstructions from DLG compared to invGrad and iDLG, while the practical performance of DLG (expectedly) is quite a lot worse than invGrad and iDLG. If the authors want to claim this is due to average vs best-case performance, they should provide more empirical evidence than Figure 9c for these discrepancies as they are noticeable in **almost** all figures in the paper.
- **Applicability of the proposed bounds:**
For the proposed bounds to be computable, one needs to execute the leakage attacks on federated models and losses that jointly satisfy both $\mu$-convexity and $L$-smoothness at the same time. Unfortunately, this restricts the usability of the bound to the federated learning models and losses that are jointly "close to" representing a quadratic function, even though the original attacks are applicable and tested on much more complex models. Further, those assumptions cannot be trivially disregarded, as the bound does not only make these assumptions but requires estimates of $\mu$ and $L$ to be computed. This forces the authors to restrict their federated models in their experiments to only Logistic Regression and 2-layer convolutional neural network without activations.
Similarly, the bound also depends on upper bounds on the variance and size of gradients, which the authors are forced to unsoundly approximate even for the simple networks used.
Finally, the precision of the second term of the bound heavily depends on the ability to accurately estimate the Lipshitz constant of $\mathcal{R}$. This naturally means that more complex methods $\mathcal{R}$, such as very deep neural networks, are penalized more heavily in their second term when they should not necessarily have to be. For example, methods like [4] and [5] have been shown to be very effective at recovering user data, even if they would likely have large Lipshitz constant estimates.
- **Not important:**
1. The authors should cite [1] and [3] as prior frameworks that attempt to analyze leakage attacks.
2. Some attacks on language models attacks like [6] and [7], might be hard to represent in this framework due to being more look-up-based than optimization-based. Similarly, some malicious server attacks might be hard to represent in this framework due to their dependency on the particular malicious weights sent to the client, which are far away from $w^\*$ like in [8]. Approaches like [9] also cannot be handled. To this end, the paper can benefit from a discussion of the limitations of the bound in terms of what types of leakage attacks it supports.
3. In Sec. 3.2, the definition of $\mathcal{R}(w_t)$ has expectation over $(x,y)$ which makes no sense in this context.
4. In Sec 3.1, the authors claim that full device participation is "unrealistic in practice". Due to cross-silo applications of FL, they might want to tune this claim down.
5. In Algorithm 2, Line 1, second statement: $\phi_h$ should be $\phi_H$ instead
5. In Appendix C.1.1, the notation for the regularizer parameter of the Logistic Regression $\gamma$ clashes with $\gamma$ used in the various bounds in the paper.
6. In Appendix C.1.1, the paragraph on computing $L$ provides two different bounds on $L$ - one with and one without $2\gamma$

**Questions:**

- Can you comment on why the bound is important if it requires estimating $\mathcal{R}(w^\*)$?
- Can you comment on the discrepancy between practical and bound behavior of leakage attacks with respect to $t$?
- In your experiments, when you estimate $[x-\mathcal{R}(w^\*)]^2$ what do you average across?
- Can the authors explain the appropriate details of their unrolled network, its parameters, and training?
- Can the authors update the paper to fix the scale of empirical results?
- Can the authors provide correlation metrics between bound and empirical results?
- Can the authors experiment with the effect of the network and attack hyperparameters on their bound?
- Can the authors provide more experiments like Figure 9c?
- Can the authors provide heterogeneity experiments?

All in all, I feel that the tackled problem is important, and using theoretical bounds is a promising way of tackling it. I just find the particular bound oversimplistic in the way it handles the leakage attacks over time and not that useful due to its dependence on $\mathcal{R}(w^\*)$. I think these limitations lead to two major issues - one is that the bound does not model the practical properties of the gradient leakage attacks analyzed that well, and the second is that the bound is as hard, if not harder, to apply than the existing rudimentary evaluation techniques used in papers. On top of that, the authors omit several very important details about their experimental setup and make the empirical plots hard to read by choosing a bad scaling factor, making it hard to feel convinced by some of the experiments provided. Note that I have given you a 3, as I am not allowed to give you a 4 (due to ICLR rules this year), but I am leaning towards a 4.

[1] Balunović, Mislav, et al. "Bayesian framework for gradient leakage." arXiv preprint arXiv:2111.04706 (2021).
[2] Dimitrov, Dimitar Iliev, et al. "Data leakage in federated averaging." Transactions on Machine Learning Research (2022).
[3] Wei, Wenqi, et al. "A framework for evaluating gradient leakage attacks in federated learning." arXiv preprint arXiv:2004.10397 (2020).
[4] Wu, Ruihan, et al. "Learning to invert: Simple adaptive attacks for gradient inversion in federated learning." Uncertainty in Artificial Intelligence. PMLR, 2023.
[5] Dongyun Xue, Haomiao Yang, Mengyu Ge, Jingwei Li, Guowen Xu, and Hongwei Li. Fast generation-based gradient leakage attacks against highly compressed gradients. IEEE INFO316 COM 2023 - IEEE Conference on Computer Communications, 2023.
[6] Fowl, Liam, et al. "Decepticons: Corrupted transformers breach privacy in federated learning for language models." arXiv preprint arXiv:2201.12675 (2022).
[7] Gupta, Samyak, et al. "Recovering private text in federated learning of language models." Advances in Neural Information Processing Systems 35 (2022): 8130-8143.
[8] Wen, Yuxin, et al. "Fishing for user data in large-batch federated learning via gradient magnification." arXiv preprint arXiv:2202.00580 (2022).
[9] Zhu, Junyi, and Matthew Blaschko. "R-gap: Recursive gradient attack on privacy." arXiv preprint arXiv:2010.07733 (2020).

**Details Of Ethics Concerns:**

Not needed

---

> ### Author Response · Authors · 2023-11-20
> **Response to the Reviewer - Part I**
>
> We thank the reviewer for the constructive comments and suggestions!
>
> **Comment#1**: Why is the error bound important if it requires estimating $\mathcal{R}(w^*)$)**?  The first term of the bound still depends on the performance at $w^*$ and the leakage algorithm $\mathcal{R}$. Presence of randomness (e.g., from the initialization) also remains as hard as before.
>
> **Response:** First, under a given *convex* FL algorithm, the optimal $w^*$ is unique (and can be obtained), and *all* leakage algorithms are compared using the same optimal model. We observe that, based on this optimal model, all leakage algorithms have relatively stable (variance less than 10) and small reconstruction errors of the first term (though the presence of randomness, e.g., from the initialization). Instead, the existing attacks are tested on intermediate $w^t$, which could cause unstable empirical reconstruction errors, as shown in Figure 1.
>
> Second, we also noticed that the second term largely dominates the error bound, compared with the first one.  For instance, in the default setting on MNIST, the two terms in DLG, iDLG, and invGrad, are (30.48, 396.72), (25.25, 341.10), (22.06, 218.28), respectively.
>
> We admit that *convex* loss may not be satisfied in deep neural networks and generalizing our theoretical results to non-convex losses is important future work.
>
> **Comment#2:** Discrepancy between practical and bound behavior of leakage attacks with respect to $t$ (both practically and theoretically, the bound predicts that as $t$ increases, the vulnerability of the attacked model increases. This is in stark contradiction with the empirical observations about gradient leakage attacks where exactly the opposite is true)
>
> **Response:** In most of the results, we found the best empirical errors are very stable with respect to $t$. I guess one possible reason might be that the models used to train the FL are relatively simple (i.e., convex shallow models) and the trend of the model $w^t$ has a certain pattern (e.g., it gradually converges to the optimal model $w^*$). Instead, the existing attacks use nonconvex deep neural networks to train FL, which could memorize the data in the first few iterations, and hence they show less empirical reconstruction error with a small $t$.
>
> Our theoretical results show the error bound decreases w.r.t. $t$. As pointed out by the reviewer, this is due to the error bound in the second term decreasing as $t$ increases.
>
> **Comment#3: Unclear or missing implementation details**
>
> **Comment#3A:** what is the average taken over in $\mathbb{E}[x-\mathcal{R}(w^*)]$
>
> **Response:** The average is over the randomness in $\mathcal{R}$, e.g., different initializations.
>
> **Comment#3B:** How is $\Gamma$ approximated in practice? How is heterogeneity controlled in general in the experiments provided? Can the authors provide heterogeneity experiments?
>
> **Response:** Recall $\Gamma = \mathcal{L}^* - p_k \mathcal{L}_k^*$, where $p_k$ can be understood as the fraction of data samples in client $k$ over all clients' data. $\mathcal{L}^*$ is global loss, and $\mathcal{L}_k^*$ is client $k$’s loss, both of which can be calculated under the convex loss setting. In our results, *the heterogeneity can be controlled by the number of classes per client*. For instance, the clients that have 1 class of data samples show a larger level of heterogeneity than the clients that have 2 classes of data samples—as the client losses in the former case could be more diverse than those in the latter case.
>
> Figure 9(a) demonstrates the results on different levels of heterogeneity. We can see a large level of heterogeneity makes the attack easier, i.e., a lower error bound or empirical error. This is consistent with the existing literature.
>
>
> **Comment#3C:** Where are parameters $\theta^i$ coming from in Figure 2 and Algorithm 1?  Can you elaborate on why you tune them with layer-wise methods instead of SGD?
>
> **Response:** Sorry for the confusion. Each $\theta^i$ means the model parameter connecting the $i$-th layer and $i+1$th layer in the unrolled deep neural network. In our context,  each  $i$-th layer is the intermediate reconstructions $x_i’$ generated by the attack algorithm $\mathcal{R}$. The unrolled deep neural network is to learn $\theta^i$’s via reproducing the intermediate reconstructions $x_i’$. More training details are given in the Appendix C.1.1.
>
> The reason to train the network in a layer-wise fashion is because the number of model parameters is large (e.g., the \#parameter in each layer is the \#pixels * \#pixels). However, we use SGD to update the parameters within each layer-wise training.

---

> ### Author Response · Authors · 2023-11-20
> **Response to the Reviewer - Part II**
>
> **Comment#4: Problems with the presented evaluation**
>
> **Comment#4A: Use two different scales for the bounds and the empirical results**
>
> **Response:** Thanks for the suggestion. Will do.
>
> **Comment#4B:** the authors do not provide a correlation metric between their bound and the empirical evaluation results. Can the authors provide correlation metrics between bound and empirical results?
>
> **Response:** Our theoretical results show that the *one-snapshot* empirical errors may not have a strong relation with the bounded errors. This is because *one-snapshot* empirical errors obtained by the existing attack can be significantly different under different initializations (as verified in Figure 1).  However, Figure 9c shows that the error bound per attack has a *strong correlation* with its empirical errors *in expectation*. We tested that with the 10 initializations, the Pearson correlation coefficient between the error bound and the averaged empirical errors on the four attacks are larger than 0.9.
>
> Our theoretical results (in Theorems 1 and 2) show that, when an attacker’s reconstruction function has a *smaller Lipschitz* constant (in the second term), then this attack intrinsically performs better, i.e., more gradient leakage. Note that the second term is often much larger than the first term in the error bound.
>
> **Comment#4C:** More error bounds on the same gradient leakage attack with different models and architectures. More evaluations across different hyperparameters such as initialization strategies, regularizer strengths.
>
> **Response:** The main motivation of the paper is to *theoretically compare the existing gradient leakage attacks with a unified framework*. This provides guidance to compare attacks more fairly and reasonably, as the literature only shows the comparison results at once and in different settings.
>
> We also note the existing attacks do use different models/architectures, and our takeaway is that the Lipschitz of the model/attack algorithm largely determines the attack performance.
>
> In addition, we have tested our error bound with different initializations and we found it is insensitive to them. This is because the unrolled deep neural network learns the intrinsic Lipschitz of the attack algorithm $\mathcal{R}$.
>
>
> **Comment#4D:** All experiments should show average behavior for the author's claims to be substantiated (Can the authors provide more experiments like Figure 9c?)
>
> **Response:** Thanks for your suggestion! The reason we show the best empirical results is to replicate the results shown in the literature. We will add average empirical results as well as the Pearson correlation coefficients.

---

> ### Comment · Reviewer_sB6V · 2023-11-21
> **Response to Response to the Reviewer sB6V**
>
> The reviewer feels this paper is suffering from the problem that even its authors do not know what they want to achieve with their proposed bound.  I think this is reflected in all reviewers' scores. I see three options:
> - A practical bound used to evaluate practical gradient leakage algorithms. For this, the authors should show that their bound reflects real applications - e.g. the bound is either faster to compute than current evaluations or allows computing of optimal hyperparameters/initialization of the attack. The reviewer feels the current bound is hopeless in achieving both of these things because it is independent of many of the useful attack hyperparameters, it is computationally expensive, and requires unrealistic models to be computed.
> - A theoretical bound that closely predicts how the attack behaves on average with respect to a list of parameters of the attack. This can potentially be useful to make generic statements regarding what properties of one attack make it better in one setting or another. The reviewer thinks this is, again, not possible. For this, at least trends need to match, and the time/communication rounds trend presented just does not match real behavior, even by the authors' own admission.
> - A theoretical bound that represents worst-case information leakage, which can inform a client how exposed their data is.  Here, the fact you use an upper bound on the MSE is problematic.
>
> I will encourage the authors to define in which of these three settings they think their bound operates or define a new one in which the bound application is clear. I also encourage them to clearly outline it in the introduction/abstract of their paper in the next revision.
>
> **Comment#1:**  I understand that relaxing convexity as an assumption can be challenging, and therefore left for future work, but I think the authors agree that this is the ultimate aim of the bound, as presented. However, the techniques used in the proposed bound assume convexity so heavily ( by the author's own admission in non-convex settings, there is even multiple $w^*$ ) that I do not see the used techniques capable of being adapted to a non-convex setting. Further, the statement by the authors above "we should require FL training to converge to the optimal model" is also misguided. If one wants to evaluate privacy risks which is the ultimate goal of gradient leakage attacks the convexity of models and their theoretical or practical convergence properties are immaterial. What matters is whether they get deployed in realistic scenarios or not.
>
> Finally, having $\mathbb{E}||x - \mathcal{R}(w^*)||^2$,as one of your boundary terms and even more so being the smaller term of this said bound, is very counterintuitive when you think about the overall problem here. If a gradient leakage attack already achieves certain MSE in expectation on $w^*$ this is more than good enough lower bound on the privacy lost in an FL process. The rest of the terms ultimately would not matter from a defense point of view. The only reason a client cares about attacks at $w^t$ is that in practice they are stronger. This is not captured by the bound of the authors.
>
> **Comment#2:** The authors acknowledge that even for their simple models the empirical bound does not decrease with $T$. Their bound does. This suggests their bound is non-predictive of the real behaviour of these attacks. I have stated before this is due to too large overapproximations made by the bound at multiple points, as it is sound. This argues against its use as a theoretical bound used for theoretical analysis
>
> **Comment#3A:** Do you assume fixed hyperparameters for the attacks and how do you choose them?
>
> **Comment#3B:** Do you mean to say Figure 9 (a) attacks clients with different numbers of classes from MNIST, when the full federated training is still carried out on full MNIST? This needs to be better explained in the paper.
>
> **Comment#3C:** I do not understand the provided explanation. Unless I am missing something, each attack doesn't depend on any parameters $\theta_i$. Each $x_i$ is just produced from the previous $x_{i-1}$ by executing for example a single SGD step in the case of InvGrad. Why do you need to optimize any parameters in this scheme? How does the optimized parameter relate to the standard InvGrad?
>
> **Comment#4A&D:** The authors need to understand that they cannot just promise those results and expect to be accepted on the basis of the current ones. Both of these points are crucial to understand the experimental results they present. Especially for 4A this should be trivial to do and be in a revision of the paper already.
>
> All in all, my worry about the paper's main issues has deepened not resolved after reading the author's responses

---

> > ### Author Response · Authors · 2023-11-22
> > **Second Round Response to the Reviewer - Part I**
> >
> > Thanks for your further comments!
> >
> > We would like to first highlight the contributions of the paper. First, we provide a way to theoretically understand and compare the existing data reconstruction attacks, and this is the first method to do so. The underlying reason lies in that the one-snapshot/best empirical errors are not sufficient and cannot reflect the attack’s intrinsic effectiveness. Though our theoretical result focuses on simple convex FL models, it is still meaningful. This is because FL has been widely studied in applications with constrained resources (e.g., edge computing) and with insufficient data (this is actually one reason that brings the FL out). In these scenarios, the FL model is simple. For instance, the original FL [McMahan et al 2017] uses a 2-layer NNs, and we show theoretical results on the convex 2-layer ReLU networks (Pilanci & Ergen, 2020).
> >
> > Second, our theoretical results can guide the future design of theoretically better data reconstruction attacks and effective defenses. From theorems 1 and 2, a lower bounded error requires a smaller Lipschitz constant of the reconstruction function. Hence, future works can focus on designing reconstruction functions with small Lipschitz constants, while effective defenses should produce large ones.
> >
> > **Comment#0**: theoretical bound that represents worst-case information leakage, which can inform a client how exposed their data is. Here, the fact you use an upper bound on the MSE is problematic.
> >
> > **Response:** We respectfully disagree with this. The upper bounded error indicates the worst-case error of the attacks, which can be used as the metric for theoretical comparison. If attack A consistently has a smaller bounded error than attack B, it implies A theoretically outperforms B.
> >
> > **Comment#1A:** If one wants to evaluate privacy risks which is the ultimate goal of gradient leakage attacks. What matters is whether they get deployed in realistic scenarios or not.
> >
> > **Response:** I agree that, in practice, an attacker may care more about the empirical attack performance, but the key question is how to ensure the attack be effective enough? Our theoretical results could provide such a strategy: if the attack can ensure its reconstruction procedure has better smoothness (characterized by a smaller Lipschitz constant), the empirical attack effectiveness could be better theoretically.
> >
> > Also, convex FL models could be widely deployed in applications with constrained resources or/and insufficient data. It is hence still meaningful to theoretically understand the data reconstruction attacks under the simple (convex) models.
> >
> >
> > **Comment#1B**: If a gradient leakage attack already achieves certain MSE in expectation on $w^*$, this is more than good enough lower bound on the privacy lost in an FL process. The rest of the terms ultimately would not matter from a defense point of view.
> >
> > **Response:** I believe this “counterintuitive” phenomenon is from the convex FL models, but not from the theoretical result. In practice, the attacker may perform the attack at any stage of the FL training since s/he does not know which iteration enables the best attack. We hence think it is still useful to bound the error at any iteration $t$.
> >
> > **Comment#2**: The authors acknowledge that even for their simple models the empirical bound does not decrease with $t$, but their bound does. This suggests their bound is non-predictive of the real behavior of these attacks.
> >
> > **Response**: This is a misunderstanding! The conclusion is that the *best* empirical error does not decrease with $t$. For the average empirical error, it indeed decreases with $t$ and has a strong correlation with the error bound (e.g., as replied in the previous rebuttal, the Pearson correlation coefficients are larger than 0.9 on FMNIST).
> >
> > **Comment#3A**: Do you assume fixed hyperparameters for the attacks and how do you choose them?
> >
> > **Response:** One key hyperparameter is the initial input $x_0$ that is parameterized by, e.g., a mean and standard deviation of a Gaussian distribution in some attacks. We try different means and standard deviations to initialize the input. The attack then takes the initial input  (and the trained FL global model) as input and iteratively outputs the reconstruction. The attack optimization process is based on the source code and we tune the learning rate to achieve the best possible reconstruction for each initial input.
> >
> > **Comment#3B**: Do you mean to say Figure 9 (a) attacks clients with different numbers of classes from MNIST, when the full federated training is still carried out on full MNIST? This needs to be better explained in the paper.
> >
> > **Response:** Yes it is! I will add more clarifications.

---

> > > ### Author Response · Authors · 2023-11-22
> > > **Second Round Response to the Reviewer - Part II**
> > >
> > > **Comment#3C**: I do not understand the provided explanation. Unless I am missing something, each attack doesn't depend on any parameters $\theta_i$. Each $x_i$ is just produced from the previous $x_{i-1}$ by executing for example a single SGD step in the case of InvGrad. Why do you need to optimize any parameters in this scheme? How does the optimized parameter relate to the standard InvGrad?
> > >
> > > **Response:** No, the attack does depend on the parameters $\{\theta_i\}$. Let us provide more clarifications below.
> > >
> > > Imagine InvGrad is a complicated optimization-based attack aiming to iteratively reconstruct the input from a random initial input. How can you understand such attack optimization trajectory? Empirically, yes, we can generate an initial input and see the reconstructed output by InvGrad. We can repeat such a process many times until finding the expected reconstruction. This is exactly what the existing attacks do. However, the underlying attack mechanism is still unclear, making the fair comparison of existing attacks hard. For instance, attack A may happen to find an expected reconstruction (e.g., MSE=2)  using 10 random inputs, while attack B cannot find an expected reconstruction MSE<=2 within 10 random inputs, but uses 20 random inputs to reach a reconstruction MSE is 1.5. How to compare them? Some papers report A is better, while some others report B.
> > >
> > > We hence propose to theoretically understand the attack, where we map the attack optimization trajectory as learning a corresponding unrolled deep feedforward network—its hidden layers are the intermediate reconstructions ($x_{i-1}$, $x_i$, etc.) during the attack optimization, and the weight parameters are $\{\theta_i\}$. By learning these $\{\theta_i\}$’s, we can determine the upper bound Lipschitz of the unrolled feedforward network, and hence the attack algorithm, where the Lipschitz constant reflects the inherent attack effectiveness of the attack (i.e., smaller Lipschitz implies better attack).  As different attacks generate different intermediate reconstructions, the corresponding learnt $\{\theta_i\}$’s are different, so do the corresponding estimated upper bound Lipschitz of the attacks.
> > >
> > >
> > > **Comment#4A&D**: The authors need to understand that they cannot just promise those results and expect to be accepted on the basis of the current ones. Both of these points are crucial to understand the experimental results they present. Especially for 4A this should be trivial to do and be in a revision of the paper already.
> > >
> > > **Response:** We tried plotting the figures using two different scales but they do not look good. We put numbers in Figure b(c) below for reference.
> > >
> > > | $T$  | 50 | 75 | 100|  125| 150 | 175 | 200 |
> > > |:------------|--------------:|:-------------:|:-------------:|:-------------:|-------------:|-------------:|-------------:|
> > > | DLG error bound        | 1000.526  |  934.126      |  891.465 |  842.978 | 798.4475 | 700.917 | 699.3865
> > > | iDLG-error bound      | 932.16    |   858.4156     | 800.456  | 721.9732|  686.1212 | 630.2692  | 594.4172
> > > | InvGrad-error bound  | 856.2613    |   798.265    | 724.156 | 680.7888 | 624.7361 | 578.6835  | 532.6308
> > > | GGL-error bound     | 115.165    |  108.0156     | 100.446  | 96.6156 | 92.6516| 90.65416 | 89.1256
> > > | | | | | | | |
> > > | DLG-optimal error        | 45.879  |  45.3365      |  45.165 |  44.74617 | 44.389 | 44.03 | 43.6751
> > > | iDLG-optimal error      | 40.468     |   40.462    | 40.466 | 40.458| 40.455 | 40.463| 40.464
> > > | InvGrad-optimal error      | 38.7816    |  38.0489      | 37.4156 | 36.716 | 36.433| 36.05 | 35.467
> > > | GGL-optimal error      | 27.78|  27.76      | 27.4666  | 27.317 | 27.159| 27.0005 | 26.844

---

> > > > ### Comment · Reviewer_sB6V · 2023-11-22
> > > > **Re: Second Round Response to the Reviewer**
> > > >
> > > > **Comment#4A&D:** I think the provided Table aptly demonstrates my points. That is to say, when the results are provided in such a form, actual trends are noticeable in the experiment when in the original figures that was not the case. I do not understand what the possible issue of the authors was to do this visually, but I am fine with the tabular version of the results as well. Further, the table also demonstrates my point that **all** experiments need to be redone for the average case. Otherwise, the experiments just do not convey the author's intent about their bound. My point here stands that a resubmission with the full set of the results reran in that matter is required for acceptance.
> > > >
> > > > **Comment#3C:** I am an expert in this field and yet I find the explanation provided of what $\theta_i$ are w.r.t. the used algorithms very confusing. As it seems crucial for understanding the author's proposed method, I suggest the authors dedicate more space to this in the main paper. I think they also should demonstrate an example of this either for GGL or InvGrad (preferably both given how different those two methods are) in the appendix. Something like: "Two iterations of InvGrad with initialization $x_0$ correspond to ..." would be ideal.
> > > >
> > > > **Comment#3A:** I, in particular, was interested in the rest of the hyperparameters. I understand that you average the initializations.
> > > >
> > > > **Comment#2:** The newly obtained result is interesting and valuable. That said, they were only obtained after several rounds of communication with the authors, and they only constitute one of 20 figures that need the proposed changes in response to **Comment#4A&D**. Additionally, I would suggest in the next iteration of the paper the authors provide some of the explanations in this discussion as to why the average behavior in the convex case differs from practical observations.
> > > >
> > > > **Comment#0&1A&1B:** My point is simply that gradient attacks are rarely the ultimate goal of this kind of research. From my point of view, gradient inversion attacks, instead, are mostly a practical proxy for measuring the privacy promise of FL. In that context, my comments stand.
> > > >
> > > > **Comment#0:** Note that I am not saying a lower bound is the only way for a bound in the space to make sense. It is only one of the possible options I listed above. All in all, despite the authors evading my question, I think the paper lands closest to "A theoretical bound that closely predicts how the attack behaves on average with respect to a list of parameters of the attack." in my list above. But this selling point needs to be clearly expressed in the paper. The paper should provide, in this scenario, much better explanation of the predictive power of their bound for all parameters both experimentally and practically and should explain what difference practically one can observe between convex and non-convex models, which from this discussion, clearly exists. That can also be used to inform the future work in this space.
> > > > Side note and suggestion for future work: analyzing the standard deviation of the methods is also important - e.g if some can often produce lucky runs, this is also useful to analyze and know.
> > > >
> > > > All in all, I think the paper cannot be accepted like it is right now. It requires applying Comment 4A and 4D everywhere. I also think it requires a serious rewrite to focus more on explaining the significance of the author's results. I have updated my score to 5, but I will not recommend acceptance.

---

### Official Review · Reviewer_3bse · 2023-10-25

**Soundness:** 2 fair
**Presentation:** 3 good
**Contribution:** 3 good
**Rating:** 5
**Confidence:** 4

**Summary:**

This paper investigates optimization-based model inversion attacks in federated learning from a theoretical standpoint. Under specific assumptions, it determines the upper bound error for data reconstruction. Notably, the study maps the iterative algorithm used for these attacks to an unrolled deep feed-forward network. This mapping facilitates the computation of the upper bound Lipschitz Constant for the attack function using the AutoLip and Power methods. Experimental results further substantiate the presented theoretical findings.

**Strengths:**

- The paper is well-written and easy to follow.
- By calculating the upper bound error for attacks, it offers a valuable metric to assess the privacy leakage associated with model inversion attacks.

**Weaknesses:**

- The main concern is the low correlation between the proposed upper bound Lipschitz Constant and true reconstruction error. Offering a clearer and more understandable explanation would elevate the persuasiveness and lucidity of the research.
- While MSE emphasizes pixel-wise differences, this can sometimes be misleading. For instance, a reconstruction might visually appear impeccable, but if there are a few pixels that greatly deviate from their counterparts, the MSE value could be disproportionately high. Metrics like SSIM provide a perception-based assessment, and LPIPS offers a learned perceptual similarity, which might capture human perceptual judgment more accurately. It would be beneficial to integrate these non-norm-based metrics.

**Questions:**

Please see weakness section.

---

> ### Author Response · Authors · 2023-11-20
> **Response to the Reviewer**
>
> We thank the reviewer for appreciating the novelty and the constructive comments!
>
>
> **Comment#1:** low correlation between the proposed upper bound Lipschitz Constant and true reconstruction error.
>
> **Response:** This may be a misunderstanding! Our results show that the best *one-snapshot empirical errors are not consistent with the bounded errors in some cases* for certain attacks (e.g., InvGrad on FMNUST). However,  the error bound per attack indeed has a *strong correlation with its empirical errors in expectation*. See Figure 9c, where we can see the consistency between the error bounds and average empirical errors.
>
>
> **Comment#2:** MSE emphasizes pixel-wise differences are insufficient. It would be beneficial to integrate these non-norm-based metrics such as  SSIM and LPIPS.
>
> **Response:** We agree that structure based SSIM and LPIPS could provide more accurate human perceptual judgment! However, we emphasize *this is the first paper to theoretically understand the data reconstruction attack*, and we first focus on norm-based metric. Nevertheless, we acknowledge that generalizing our theoretical framework to SSIM or LPIPS face a fundamental challenge: these metrics do NOT even have an analytic form, making the error/similarity bound formulation infeasible.

---

### Official Review · Reviewer_kviX · 2023-10-30

**Soundness:** 2 fair
**Presentation:** 3 good
**Contribution:** 1 poor
**Rating:** 3
**Confidence:** 4

**Summary:**

This paper studies the data reconstruction attack in federated learning. It proposes a theoretical upper bound for the data reconstruction attack and then introduces how to calculate the upper bound. In the experiment section, the paper shows the comparison between different data reconstruction attacks and their upper bound.

**Strengths:**

1. The clarity is good. The flow of the writing is smooth.
2. In the introduction section, the paper points out three meaningful limitations of existing attack methods: high sensitivity to the initialization; evaluation highly depending on the choice of model snapshot; and lack of theoretical analysis.

**Weaknesses:**

My main concern is about the significance. In details
1. Theorems on the attack upper bound need unrealistic assumptions.
-  The FL model is required to be strongly convex and smooth, etc. This is far away from the scenario where those attack methods are located -- those attack methods are commonly evaluated for deep neural networks as the input is the image in the evaluation.
- The reconstruction method is assumed to have a Lipschitz constant. However, the attack methods studied in the paper are optimization-based attacks and do not have an explicit Lipschitz constant. Although the paper proposes a method to approximately calculate the constant, there's no guarantee how large the approximation error would be. This might introduce an additional arbitrarily large noise to calculate the upper bound.
2. As shown in the empirical results, there is a large gap between the upper bound and the attack performance. It might be hard to indicate a useful conclusion from this bound.

**Questions:**

Please see the weakness above.

---

> ### Author Response · Authors · 2023-11-20
> **Response to the Reviewer**
>
> We thank the reviewer for the constructive comments!
>
> **Comment#1:** The FL model is required to be strongly convex and smooth, while nonlinear deep neural networks are commonly used in FL.
>
> **Response:** We emphasize that this is the first work to understand the data reconstruction attack from the theoretical perspective. We note that it is challenging to bound the error in practical FL settings, e.g., non-IID data across clients. To make the problem tractable, we should require FL training to converge to the optimal model, which needs the loss to be convex.
>
> Although deep neural networks do not satisfy this assumption, FL has been also used in convex settings. For instance, the original FL [McMahan et al 2017] uses, e.g., 2-layer ReLu NNs, and we also show theoretical results on the convex 2-layer ReLU networks (Pilanci & Ergen, 2020).  We acknowledge it is important future work to generalize our theoretical results to more challenging non-convex losses.
>
> **Comment#2:** The paper proposes a method to approximately calculate the Lipschitz  constant, but there's no guarantee how large the approximation error would be.
>
> **Response:** It is challenging to bound the approximation error of the estimated Lipschitz  constant to the exact one of a *highly nonconvex* network used in the attacks. Also, our goal in the paper is to design a feasible algorithm to estimate the Lipschitz constant. We admit a better estimation algorithm can lead to a tighter upper bounded Lipschitz constant, which is an interesting future work, but not the focus of the paper. Our results show the estimated Lipschitz constant is stable.
>
>
> **Comment#3:** large gap between the upper bound and the attack performance
>
> **Response:** Note that the derived error bounds consider all randomness in FL training and data reconstruction, and hence they are the worst-case error under such randomness.  The empirical attack performance of iDLG (Zhao et al., 2020) and DLG (Zhu et al., 2019) are significantly influenced by initial parameters. As shown in Figure 1, the MSE of DLG could be very large (more than 600), which is close to the upper bound (about 700) in this case. Also shown in Figure 6 and Figure 7, GGL exhibits close upper bound and empirical error.

---

> > ### Comment · Reviewer_kviX · 2023-11-23
> > **Official Comment by Reviewer kviX**
> >
> > I have read the response and find it doesn't solve my concern on the significance in the aspects of the application scenario and the empirical findings, so I decided to keep my score.

---

### Meta-Review · Area_Chair_8h9v · 2023-12-08

**Metareview:**

This paper theoretically analyzes data reconstruction leakage in FL based on Lipschitz constant of the reconstruction function. While reviewers appreciated the rigor of the theoretical analysis, they also raised several concerns about significance of the result. Namely, the analysis requires that the FL model is strongly convex and smooth, which precludes most common FL applications. There is also a weak correlation between the theoretical bound and the true attack performance, which raises questions about whether the bound based on Lipschitz constant truly reflects attack performance. AC agrees with reviewer assessment and believes the paper is not ready for publication at this time.

**Justification For Why Not Higher Score:**

The derived bound is relies on unrealistic assumptions and does not correlate well with empirical attack performance. Reviewers unanimously voted for rejection based on these weaknesses.

**Justification For Why Not Lower Score:**

N/A

---

### Decision · Program_Chairs · 2024-01-16

Reject